# Learning Efficient Surrogate Dynamic Models with Graph Spline Networks

**Chuanbo Hua**[*]
KAIST
DiffeqML, AI4CO

**Federico Berto**[*]
KAIST
DiffeqML, AI4CO

**Michael Poli**
Stanford University
DiffeqML

**Stefano Massaroli**
Mila and University of Montreal
DiffeqML

**Jinkyoo Park**
KAIST
OMELET

## Abstract

While complex simulations of physical systems have been widely used in engineering and scientific computing, lowering their often prohibitive computational requirements has only recently been tackled by deep learning approaches. In this paper, we present GRAPHSPLINENETS, a novel deep-learning method to speed up the forecasting of physical systems by reducing the grid size and number of iteration steps of deep surrogate models. Our method uses two differentiable orthogonal spline collocation methods to efficiently predict response at any location in time and space. Additionally, we introduce an adaptive collocation strategy in space to prioritize sampling from the most important regions. GRAPHSPLINENETS improve the accuracy-speedup tradeoff in forecasting various dynamical systems with increasing complexity, including the heat equation, damped wave propagation, Navier-Stokes equations, and real-world ocean currents in both regular and irregular domains.

## 1 Introduction

For a growing variety of fields, simulations of partial differential equations (PDEs) representing physical processes are an essential tool. PDE–based simulators have been widely employed in a range of practical applications, spanning from astrophysics (Mücke et al., 2000) to biology (Quarteroni and Veneziani, 2003), engineering (Wu and Porté-Agel, 2011), finance, (Marriott et al., 2015) or weather forecasting (Bauer et al., 2015).

However, traditional solvers for physics-based simulation often need a significant amount of computational resources (Houska et al., 2012), such as solvers based on first principles and the modified Gauss-Newton methods. Traditional physics-based simulation heavily relies on knowledge of underlying physics and parameters requiring *ad-hoc* modeling, which is sensitive to design choices. Finally, even the best traditional simulators are often inaccurate due to the difficulty in approximating real dynamics (Kremer and Hancock, 2006; Oberkampf, 2019; Sanchez-Gonzalez et al., 2020).

An attractive alternative to traditional simulators is to use deep learning to train surrogate models directly from observed data. Data-driven surrogate models can generate

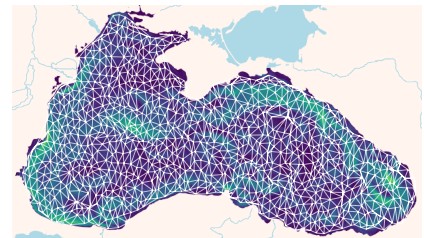

Figure 1.1: Wind speed forecasts on the Black Sea. GRAPHSPLINENETS can learn fast and accurate surrogate models for complex dynamics on irregular domains.

---

[*]Equal contribution authors.

37th Conference on Neural Information Processing Systems (NeurIPS 2023).

predictions based on the knowledge learned automatically from training without the knowledge of the underlying differential equations. Among deep learning methods, graph neural networks (GNNs) have desirable properties such as spatial equivariance and translational invariance, allowing learning representations of dynamical interactions in a generalizable manner (Pfaff et al., 2021; Bronstein et al., 2021) and on unstructured grids. Despite the benefits of these paradigms, by increasing resolution, graph-based models require significantly heavier calculations. Graph models as Poli et al. (2019); Lienen and Günnemann (2022) predict in continuous time or continuous space. However, the substantial computational overhead due to the need for iterative evaluations of a vector field over time requires considerable computation and limits their scalability (Xhonneux et al., 2020).

In this work, we propose GRAPHSPLINENETS, a novel approach that exploits the synergy between GNNs and the *orthogonal spline collocation* (OSC) method (Bialecki and Fairweather, 2001; Fairweather and Meade, 2020) to both efficiently and accurately forecast continuous responses of a target system. We use the GNNs to efficiently get the future states on coarse spatial and temporal grids and apply OSC with these coarse predictions to predict states at any location in space and time (i.e., continuous). As a result, our approach can generate high-resolution predictions faster than previous approaches, even without explicit prior knowledge of the underlying differential equation. Moreover, the end-to-end training and an adaptive sampling strategy of collocation points help the GRAPHSPLINENETS to improve prediction accuracy in continuous space and time.

We summarize our contributions as follows:

- We introduce GRAPHSPLINENETS, a novel learning framework to generate fast and accurate predictions of complex dynamical systems by employing coarse grids to predict time and space continuous responses via the differentiable orthogonal spline collocation.

- We propose an adaptive collocation sampling strategy that adjusts the locations of collocation points based on fast-changing regions, enabling dynamic improvements in forecasting accuracy.

- We show GRAPHSPLINENETS are competitive against existing methods in improving both accuracy and speed in predicting continuous complex dynamics on both simulated and real data.

## 2 Related Works

**Modeling dynamical systems with deep learning**  Deep neural networks have recently been successfully employed in accelerating dynamics forecasting, demonstrating their capabilities in predicting complex dynamics often orders of magnitude faster than traditional numerical solvers (Long et al., 2018; Li et al., 2021; Berto et al., 2022; Pathak et al., 2022; Park et al., 2022; Poli et al., 2022; Lin et al., 2023a,b; Boussif et al., 2022, 2023). However, these approaches are typically limited to structured grids and lack the ability to handle irregular grids or varying connectivity. To address this limitation, researchers have turned to graph neural networks (GNNs) as a promising alternative: GNNs enable learning directly on irregular grids and varying connectivity, making them well-suited for predicting system responses with complex geometric structures or interactions (Li et al., 2022a). Additionally, GNNs inherit physical properties derived from geometric deep learning, such as permutation and spatial equivariance (Bronstein et al., 2021), providing further advantages for modeling complex dynamics. Alet et al. (2019), one of the first approaches to model dynamics with GNNs, represent adaptively sampled points in a graph architecture to forecast continuous underlying physical processes without any *a priori* graph structure. Sanchez-Gonzalez et al. (2020) extend the GNN-based models to particle-based graph surrogate models with dynamically changing connectivity, modeling interactions through message passing. Graph neural networks have also recently been applied to large-scale weather predictions (Keisler, 2022). However, GNNs are limited in scaling in both temporal and spatial resolutions with finer grids as the number of nodes grows.

**Accelerating graph-based surrogate models**  An active research trend focuses on improving the scalability of graph-based surrogate models. Pfaff et al. (2021) extend the particle-based deep models of (Sanchez-Gonzalez et al., 2020) to mesh-based ones. By employing the existing connectivity of mesh edges, Pfaff et al. (2021) demonstrate better scalability than proximity-based online graph creation modeling particle interactions. Li et al. (2022b) extend convolutional operators to irregular grids by learning a mapping from an irregular input domain to a regular grid: under some assumptions on the domain, this allows for faster inference time compared to message passing. Fortunato et al. (2022) build on the mesh-based approach in (Sanchez-Gonzalez et al., 2020) to create multi-

scale GNNs that, by operating on both coarse and fine grids, demonstrate better scalability compared to single-scale GNN models. GRAPHSPLINENETS do not need to learn additional mappings between different domains or need additional parameters in terms of different scales thus reducing computational overheads by operating directly on unstructured coarse meshes. Our method bridges the gaps between coarse and fine grids in time and space via the orthogonal spline collocation (OSC), thus improving GNN-based methods' scalability.

**Collocation methods** Collocation and interpolation methods[2] are used to estimate unknown data values from known ones (Bourke, 1999). A fast and accurate collocation method with desirable properties in modeling dynamics is the *orthogonal spline collocation* (OSC) (Bialecki, 1998; Bourke, 1999). Compared to other accurate interpolators with $O(n^3)$ time complexity, the OSC has a complexity of only $O(n^2 \log n)$ thanks to its sparse structure, allowing for fast inference; moreover, it is extremely accurate, respects $\mathcal{C}^1$ continuity, and has theoretical guarantees on convergence (Bialecki, 1998). Due to such benefits, a few research works have aimed at combining OSC with deep learning approaches. Guo et al. (2019); Brink et al. (2021) combine deep learning and collocation methods by directly learning collocation weights; this provides advantages over traditional methods since the OSC can ensure $\mathcal{C}^1$ continuity compared to mesh-based methods. A recent work Boussif et al. (2022) has studied learnable interpolators and demonstrated that it is possible to learn an implicit neural representation of a collocation function. GRAPHSPLINENETS differ from previous neural collocation methods in two ways: first, we decrease the computational overhead by not needing to learn interpolators or collocation weights; secondly, by interpolating not only in space but also in time, our method enables considerable speedups by employing coarser temporal grids.

## 3 Background

### 3.1 Problem Setup and Notation

We first introduce the necessary notation that we will use throughout the paper. Let superscripts denote time, and subscripts denote space indexes as well as other notations. We represent the state of a physical process at space location $\mathbf{X} = \{\mathbf{x}_i, i = 1, \cdots, N\}$ and time $t \in \mathbb{R}$ as $\mathbf{Y}^t = \{y_i^t, i = 1, \cdots, N\}$ where $N$ represents the number of sample points and $\Omega \subset \mathbb{R}^D$ is a $D$-dimension physical domain. A physical process in this domain can be described by a solution of PDEs, i.e., $y_i^t = u(\mathbf{x}_i, t)$. The objective of a dynamics surrogate model is to estimate future states $\{\hat{\mathbf{Y}}^t, t \in \mathbb{R}_+\}$ given the initial states $\mathbf{Y}^0$.

### 3.2 Message Passing Neural Networks

We employ a graph $\mathbf{G}^t = \{\mathbf{v}_i^t, \mathbf{e}_{ij}^t\}$ to encode the status of physical process at sample points $\mathbf{X}$ at $t$. $\mathbf{v}_i^t$ and $\mathbf{e}_{ij}^t$ denote the attribute of sample node $i$ and the attribute of the directed edge between node $i$ and $j$, respectively. Node attributes are encoded from sample points state information while edge attributes are the distance between every two nodes. *Message passing neural networks* (MPNNs) employ an encoder–processor–decoder structure to predict the states of sample points at the next timestep by giving the previous states:

$$\hat{\mathbf{Y}}^{t+1} = \underbrace{\mathcal{D}}_{\text{decoder}} \left( \underbrace{\mathcal{P}_m(\cdots(\mathcal{P}_1}_{\text{processor}}(\underbrace{\mathcal{E}(\mathbf{X}, \mathbf{Y}^t)}_{\text{encoder}}))) \right) \tag{1}$$

where $\mathcal{E}(\cdot)$ is the encoder, $\mathcal{P}_i(\cdot)$ is the $i$-th message passing layer, and $\mathcal{D}(\cdot)$ is the decoder.

### 3.3 Orthogonal Spline Collocation Method

The OSC method consists of three steps in total: (1) partitioning the input domain and selecting collocation points, (2) generating polynomials with parameters to be determined with observations, and (3) solving equations to determine the parameters.

---

[2]Collocation and interpolation are terms that are frequently used interchangeably. While interpolation is defined as obtaining unknown values from known ones, collocation is usually defined as a finite solution space satisfying equations dictated by known (collocation) points. Thus, collocation can be considered as a flexible subset of interpolation methods that satisfies certain conditions, such as $\mathcal{C}^1$ class continuity.

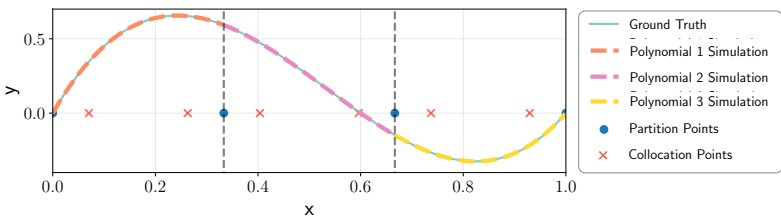

Figure 3.1: Visualization of an 1–D OSC example with the order $r = 3$ and partition number $N = 3$.

**Partitioning and initializing polynomials** The OSC aims to find a series of polynomials under order $r$ while satisfying $C^1$ continuity to approximate the solution of PDEs. To find these polynomials, we split each dimension of the domain into $N$ partitions[3][4]. Then, we initialize one $r$–order polynomial with $r + 1$ unknown coefficients in each partition. These polynomials have $N \times (r + 1)$ degrees of freedom, i.e., the variables to be specified to determine these polynomials uniquely.

**Collocation points selection and equation generation** We firstly have $2$ equations from the boundary condition and $(N - 1) \times 2$ equations from the $C^1$ continuity restriction. Then we need $N \times (r - 1)$ more equations to uniquely define the polynomials. So we select $r - 1$ collocation points where the observation is obtained to determine the parameters in each partition. By substituting the state of collocation points to the polynomials, we can get $N \times (r - 1)$ equations. Now we transfer the OSC problem to an algebraic problem. With properly selected base functions, this algebraic equation will be full rank and then has a unique solution (Lee and Micchelli, 2013).

**Solving the equations** The coefficient matrix of the generated algebraic problem is *almost block diagonal* (ABD) (De Boor and De Boor, 1978). This kind of system allows for efficient computational routines (Amodio et al., 2000), that we introduce in § 3.4. By solving the equation, we obtain the parameters for the polynomials that can be used to predict the value of any point in the domain. We illustrate an example of 1-D OSC problem in Fig. 3.1. More details and examples of the OSC are provided in Appendix A. The OSC method can also be extended to 2–D and higher-dimensional domains (Bialecki and Fairweather, 2001). In our work, we employ the 1–D and 2–D OSC approaches in the time and space domains, respectively.

## 3.4 Efficiently solving ABD matrices

Most interpolation methods need to solve linear equations. Gaussian elimination is one of the most widely used methods to solve a dense linear equation, which, however, has a $O(n^3)$ complexity (Strassen et al., 1969). Even with the best algorithms known to date, the lower bound of time complexity to solve such equations is $O(n^2 \log n)$ (Golub and Van Loan, 2013).

In the OSC method, the coefficient matrix for the linear equation follows the ABD structure, which we can efficiently solve with a time complexity $O(n^2)$ by the COLROW algorithm (Diaz et al., 1983) as shown in Fig. 3.2 (a). The core idea for this method is that by using the pivotal strategy and elimination multipliers, we can decompose the coefficient matrix into a set of a permutation matrix and upper or lower triangular matrix that can be solved in $O(n^2)$ time each. The latest package providing this algorithm is in the FORTRAN programming language: our re-implementation in PyTorch (Paszke et al., 2019) allows for optimized calculation, GPU support, and enabling the use of automatic differentiation, as shown in Fig. 3.2 (b).

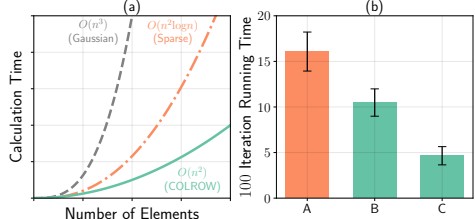

Figure 3.2: (a) The COLROW algorithm has the lowest computational complexity with $O(n^2)$ compared to the Gaussian elimination algorithm with $O(n^3)$ or lower bounds of the generic sparse solvers with $O(n^2 \log n)$. (b) Running time for 100 iterations of A: OSC+LU decomposition, B: OSC+COLROW solver, and C: OSC+COLROW solver with GPU acceleration.

---

[3]Note that these partitions do not need to be isometric, thus allowing for flexibility in their choice.

[4]We employ the Gauss-Legendre quadrature for choosing the collocation points in regular boundary problems. We further discuss the choice of collocation points in Appendix B.1.

# 4 Methodology

## 4.1 GRAPHSPLINENETS

Fig. 4.1 depicts the entire architecture of our model. GRAPHSPLINENETS can be divided into three main components: ○ message passing neural networks (MPNN), ● time-oriented collocation and ○ space-oriented collocation. The MPNN takes observations at collocation points as input and infers a sequence of discrete future predictions via autoregressive rollouts. We then use the time-oriented and space-oriented collocation methods on these discrete predictions to obtain functions that can provide both time and space-continuous predictions.

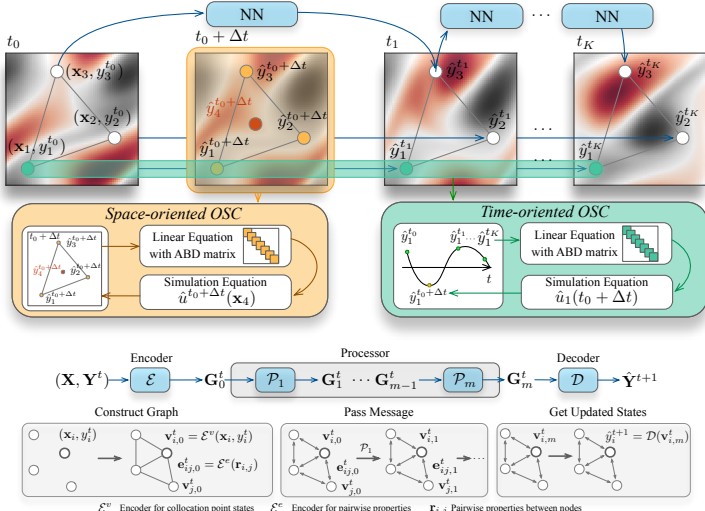

Figure 4.1: ○ Message passing neural networks take as inputs current states and employ an encoder-processor-decoder architecture for obtaining the next states autoregressively. ● time-oriented OSC obtains continuous predictions in time while ○ Space-oriented OSC is employed to obtain continuous predictions in space. The model is trained end-to-end efficiently by leveraging sparsity in the almost block diagonal (ABD) matrices of the collocation method.

**Assign collocation points** We follow the same rule to select collocation points in a 2–D domain as described in § 3.3 to select sample points $\{\mathbf{x}_i\}$ for a 2–D domain. We have the states of these sample points at the initialization time frame $\{y_i^{t_0}\}$.

○ **Message passing neural networks** The neural networks take the states of sample points at the initial time frame $\{(\mathbf{x}_i, y_i^{t_0})\}$ as input and employ an encoder-processor-decoder architecture for obtaining the next steps states $\hat{y}_i^{t_j}, j = 1, \cdots, N$ autoregressively.

● **Time-oriented OSC** For each sample point, taking $\mathbf{x}_i$ as an example, the neural networks generate a sequence of predictions along time which can be considered as a 1–D OSC problem. In this time-oriented OSC problem, collocation points are time steps $\{t_0, t_1, \cdots, t_K\}$ and values at these time steps $\{y_i^{t_0}, \hat{y}_i^{t_1}, \cdots, \hat{y}_i^{t_K}\}$, and the output is the polynomials $\hat{u}_i(t), t \in [t_0, t_K]$ which provides a continuous prediction along time; thus, we can obtain predictions at any time.

○ **Space-oriented OSC** For each time frame, we take $t_k$ as an example, the neural network generates prediction values of sample points which can be considered a 2–D OSC problem. In this space-oriented OSC problem, collocation points are positions of sample points $\{\mathbf{x}_i\}$, their values at this time frame $\{\hat{y}_i^{t_k}\}$, and the output is polynomials $\hat{u}^{t_k}(\mathbf{x}), \mathbf{x} \in \Omega$ which provides a continuous prediction in the domain. Now we get the prediction of every time frame at any position. Note that the space-oriented OSC can use the time-oriented OSC as the input, which means that we can use the space-oriented OSC at any time frame and then get a spatio-temporal continuous predictions $\hat{u}(\mathbf{x}, t), (\mathbf{x}, t) \in \Omega \times [t_0, t_K]$.

## 4.2 Training strategy and loss function

GRAPHSPLINENETS predicts the fine resolution states in a hierarchical manner. MPNN uses the sample points at the initialized state as input to propagate rollouts on these points as shown in Fig. 4.2. Based on MPNN outputs, two OSC problems are solved to find the polynomial parameters $\phi = \text{OSC}(\text{MPNN}(\mathbf{X}, \mathbf{Y}^{t_0}; \theta))$ which allow for spatiotemporal continuous outputs $\hat{u}(\mathbf{x}_i, t_k; \phi)$. Note that solving OSC problems is equivalent to predicting the future space and time continuous states, given the sample inputs.

Inherently, training the parameters, $\theta$ of the MPNN is cast as a bi-level optimization problem as $\theta^* = \arg\min_\theta \mathcal{L}(\phi, \theta)$ where $\phi = \text{OSC}(\text{MPNN}(\mathbf{X}, \mathbf{Y}^{t_0}; \theta))$. The loss function $\mathcal{L}$ is defined as:

$$L = \overbrace{\sum_{i=0}^{N} \sum_{k=0}^{K} \|y_i^{t_k} - \hat{y}_i^{t_k}\|^2}^{L_s \equiv \texttt{sample points error}} + \underbrace{\sum_{i=0}^{N_i} \sum_{k=0}^{N_t} \|y_i^{t_k} - \hat{u}(\mathbf{x}_i, t_k)\|^2}_{L_i \equiv \texttt{interpolation points error}} \quad (2)$$

The left term $L_s$ of Eq. (2) is the loss of the MPNN output $\hat{y}_i^{t_k}$, where $N$ is the number of spatial sample points, and $K$ is the number of MPNN rollout steps.

The left term $L_s$ solely depends on the MPNN parameters $\theta$. The right term $L_i$ is the prediction loss of the high-resolution interpolation points $(\mathbf{x}_i, y_i^{t_k})_{i=0, k=0}^{N_i, N_t}$ along time and space, where $N_i$ is the number of spatial interpolation points at the one-time frame, and $N_t$ is the number of interpolation time frames. Note that the space and time continuous prediction $\hat{u}(\mathbf{x}_i, t_k; \phi)$ is constructed based upon the optimized parameters $\phi = \text{OSC}(\text{MPNN}(\mathbf{X}, \mathbf{Y}^{t_0}; \theta))$, the parameters that are obtained by solving OSC problems with MPNN predictions, $\text{MPNN}(\mathbf{X}, \mathbf{Y}^{t_0}; \theta)$. Thus, both loss terms can be optimized with respect to

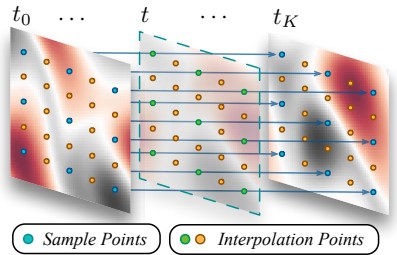

Figure 4.2: Illustration of data points used for model training. Interpolation points include ⬤ time-oriented OSC and ⬤ space-oriented OSC interpolation points. Here, $t$ can be any time frame in $(t_0, t_K)$.

the MPNN parameters $\theta$, and the whole model is trained end-to-end with automatic differentiation through the OSC. This optimization scheme is akin to Bialas and Karwan (1984); Tuy et al. (1993) that solve a bi-level optimization by flattening the loss for the upper- and lower-level problems.

## 4.3 Adaptive collocation sampling

To allow for prioritized sampling of important location regions, we dynamically optimize the positions of collocation points via their space gradients of the partial derivative values over time.

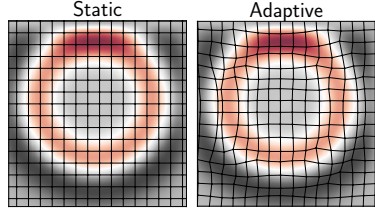

Figure 4.3: Adaptive collocation strategy: mesh points converge towards areas with faster changing regions. In this example, collocation points move towards the moving wave crests.

More specifically, after getting the continuous simulation function $\hat{u}$, the partial derivative equation of time provides the changes of the domain with time. Then, we calculate the maximum gradient vector for each collocation point to move them to more dynamic regions, i.e., with the largest change over time, which causes higher errors in the static collocation points condition. We formulate this process by

$$\tilde{\mathbf{x}}_i = \mathbf{x}_i + \beta \mathbf{v}, \mathbf{v} = \arg\max_{\mathbf{v}} \nabla_{\mathbf{v}} \frac{\partial \hat{u}}{\partial t}(\mathbf{x}_i, t_k) \quad (3)$$

where $\beta$ is the coefficient of the maximum gradient vector. We project collocation points back to the partition if the gradient step moves them outside to ensure a sufficient number of collocation points in each partition cell. We illustrate the adaptive collocation points in Fig. 4.3. We use the states at optimized positions adapted from history rollouts as the subsequent rollouts input. By adjusting the collocation points position, our model can focus on more dynamic parts of the space to get a more accurate prediction.

## 5 Experiments

### 5.1 Datasets and Training Settings

We evaluate GRAPHSPLINENETS on five dynamical systems of increasing challenge. Simulated datasets consist of three PDEs (partial differential equations), including Heat, Damped Wave, and Navier-Stokes Equations. The two empirical datasets include the Ocean and Black Sea datasets; the Black Sea introduces significant challenges in scalability in the number of nodes, complex dynamics, irregular boundary, and non-uniform meshing. All the models employ the same structure of encoder-processor-decoder for fair comparisons and the same amount of training data in each testing domain. While inputs of baseline models are directly all of the available data points, inputs of our OSC-based models are only an initialized $12 \times 12$ collocation point unitary mesh at the initial state and fewer in-time sample points. For the Black Sea dataset, we construct a mesh via Delaunay Triangulation with 5000 data points for baselines and 1000 for GRAPHSPLINENETS as done in Lienen and Günnemann (2022). More details on datasets are available in Appendix B. We value open reproducibility and make our code publicly available[5].

### 5.2 Evaluation metrics and baselines

We evaluated our model by calculating the mean square error (MSE) of rollout prediction steps with respectively ground truth for $t \in [0, 5]$ seconds. We employ relevant baselines in the field of discrete-step graph models for dynamical system predictions. *Graph convolution networks* (GCNs) (Kipf and Welling, 2016) and GCN with a hybrid *multilayer perceptron* (MLP) model are employed as baselines in the ablation study. We also compare our approach with one widely used baseline that employs linear interpolation for physics simulations allowing for continuous predictions in space, i.e., GEN (Alet et al., 2019). A similar setup considering the inherent graph structure and employing MPNNs in mesh space is used by (Pfaff et al., 2021) (MeshGraphNet, MGN in our comparisons). We utilize MGN as the MPNN building block for our GRAPHSPLINENETS.

### 5.3 Quantitative Analysis

We consider a series of ablation models on four datasets to demonstrate the effectiveness of our model components in our approach in multiple aspects. Quantitative results of ablation study models are shown in Table 5.1 [right]. The ablated models are:

- MGN: MeshGraphNet model with 3 message passing layers from Pfaff et al. (2021).
- MGN+OSC(Post): model with 3 message passing layers and only post-processing with the OSC method, i.e., we firstly train a MGN model, then we use the OSC method to collocate the prediction as a final result without end-to-end training.
- MGN+OSC: MGN with OSC-in-the-loop that allows for end-to-end training.
- MGN+OSC+Adaptive: MGN+OSC model that additionally employs our adaptive collocation point sampling strategy.

**Post processing vs end-to-end learning** We show the effectiveness of end-to-end learning architecture by comparing the MGN+OSC(Post) and MGN+OSC models. Table 5.1 shows that MGN+OSC has a more accurate prediction than MGN+OSC(Post) by more than $8\%$ percent across datasets. This can be explained by the fact that, since the OSC is applied end-to-end, the error between MGN prediction steps is backpropagated to the message passing layers. In contrast, the model has no way of considering such errors in the post-processing steps.

**Adaptive collocation points** We further show the effectiveness of the adaptive collocation strategy by comparing the MGN+OSC and MGN+OSC+Adaptive: Table 5.1 shows that MGN+OSC+Adaptive has a more accurate prediction than MGN+OSC, i.e., more than $5\%$ improvement on long rollouts in the Black Sea dataset. Adaptive collocation points encourage those points to move to the most dynamic regions in the domain, which is not only able to place

---

[5]https://github.com/kaist-silab/graphsplinenets

Table 5.1: Mean Squared Error propagation and runtime for 5 second rollouts. GRAPHSPLINENETS consistently outperform baselines in both accuracy and runtime. Smaller is better (↓). Best in **bold**; second underlined.

| Dataset | Metric | Baselines | | | | GRAPHSPLINENETS | | |
|---|---|---|---|---|---|---|---|---|
| | | GCN | GCN+MLP | GEN | MGN | MGN+OSC(Post) | MGN+OSC | MGN+OSC+Adaptive |
| **Heat Equation** | MSE ($\times 10^{-3}$) | $6.87 \pm 1.00$ | $5.02 \pm 0.89$ | $2.92 \pm 0.23$ | $3.01 \pm 0.38$ | $1.68 \pm 0.18$ | $1.14 \pm 0.11$ | **$1.07 \pm 0.28$** |
| | Runtime [s] | $3.26 \pm 0.12$ | $3.02 \pm 0.10$ | $6.87 \pm 0.10$ | $6.99 \pm 0.12$ | $1.52 \pm 0.09$ | **$1.38 \pm 0.10$** | $1.41 \pm 0.12$ |
| **Damped Wave** | MSE ($\times 10^{-1}$) | $10.5 \pm 1.65$ | $9.90 \pm 1.52$ | $6.49 \pm 0.62$ | $7.82 \pm 0.88$ | $4.98 \pm 0.29$ | $4.60 \pm 0.27$ | **$4.51 \pm 0.31$** |
| | Runtime [s] | $0.95 \pm 0.08$ | $0.82 \pm 0.07$ | $1.13 \pm 0.09$ | $1.38 \pm 0.10$ | $0.45 \pm 0.05$ | **$0.39 \pm 0.04$** | $0.42 \pm 0.09$ |
| **Navier Stokes** | MSE ($\times 10^{-1}$) | $4.24 \pm 0.95$ | $3.91 \pm 0.99$ | $3.45 \pm 0.24$ | $3.66 \pm 0.33$ | $2.58 \pm 0.28$ | $2.21 \pm 0.27$ | **$2.02 \pm 0.30$** |
| | Runtime [s] | $0.91 \pm 0.08$ | $0.88 \pm 0.07$ | $1.01 \pm 0.09$ | $1.21 \pm 0.10$ | $0.51 \pm 0.05$ | **$0.47 \pm 0.04$** | $0.49 \pm 0.09$ |
| **Ocean Currents** | MSE ($\times 10^{-1}$) | $6.02 \pm 1.12$ | $5.23 \pm 0.98$ | $4.55 \pm 0.52$ | $4.75 \pm 0.61$ | $3.82 \pm 0.49$ | $3.61 \pm 0.38$ | **$3.34 \pm 0.44$** |
| | Runtime [s] | $3.56 \pm 0.15$ | $3.38 \pm 0.13$ | $7.11 \pm 0.31$ | $7.15 \pm 0.25$ | $1.61 \pm 0.12$ | **$1.44 \pm 0.09$** | $1.57 \pm 0.11$ |
| **Black Sea** | MSE ($\times 10^{-1}$) | $7.73 \pm 0.28$ | $7.12 \pm 0.35$ | $6.22 \pm 0.31$ | $6.41 \pm 0.42$ | $4.37 \pm 0.33$ | $4.23 \pm 0.17$ | **$3.91 \pm 0.27$** |
| | Runtime [s] | $8.27 \pm 0.33$ | $8.11 \pm 0.28$ | $12.81 \pm 0.19$ | $13.35 \pm 0.27$ | $5.87 \pm 0.13$ | **$5.13 \pm 0.21$** | $5.79 \pm 0.15$ |

greater attention on hard-to-learn parts in space but also can let the OSC method implicitly develop a better representation of the domain. Empirical quantitative results on the five datasets are shown in Table 5.1. In the heat equation dataset, our approach reduces long-range prediction errors by $64\%$ with only $20\%$ of the running time compared with the best baseline model. In the damped wave dataset, our approach reduces errors by $42\%$ with a $48\%$ reduction in inference time.

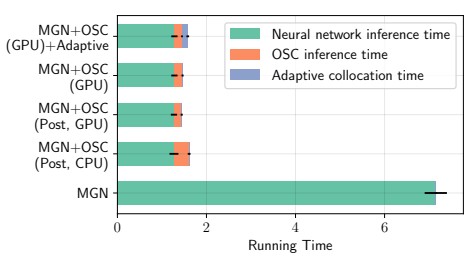

Figure 5.1: Inference time benchmark: the OSC and adaptive collocation points result in a minor overhead; the overall scheme result in a fraction of the baseline running time.

In the Navier-Stokes dataset, our method reduces $31\%$ long-range prediction errors while requiring $37\%$ less time to infer solutions compared to the strongest baseline.

**Inference times** We show the effectiveness of the COLROW algorithm in accelerating the OSC speed by comparing the OSC method with one of the most commonly used algorithms for efficiently solving linear systems[6] and the OSC with the COLROW solver in Fig. 3.2. Our differentiable OSC and adaptive collocation only result in a minor overhead over the graph neural network. The overall scheme is still a fraction of the total in terms of inference time as shown in Fig. 5.4, while making GRAPHSPLINENETS more accurate.

## 5.4 Sensitivity Analysis

**Interpolation and collocation methods** We demonstrate the efficiency of the OSC method by comparing the combination of MGN with different interpolation and collocation methods, including linear interpolation, cubic interpolation, and B-spline collocation methods. These models are implemented in the end-to-end training loop, and we use these methods in both the time and space dimensions. Results are shown in Fig. 5.2 where we measured the mean square error and running time of 3 second rollouts predictions by varying the number of collocation points from $(2 \times 2)$ to $(16 \times 16)$. The model MGN+OSC shows the best accuracy while having a shorter running time. Even though the linear interpolation can be slightly faster than the OSC, it shows a considerable error in the prediction and does not satisfy basic assumptions such as Lipschitz continuity in space.

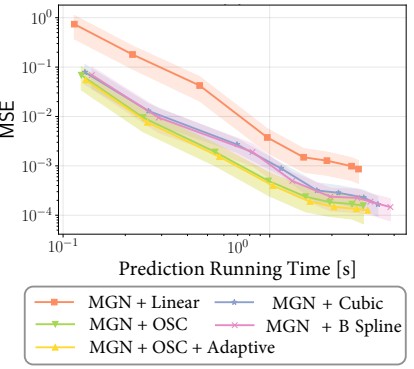

Figure 5.2: GRAPHSPLINENETS (MGN+OSC+Adaptive) is Pareto-optimal compared to combinations with other interpolation methods.

---

[6]We employ for our experiments `torch.linalg.solve`, which uses LU decomposition with partial pivoting and row interchanges. While this method is numerically stable, it still has a $O(n^3)$ complexity with a $O(n^2 \log(n))$ lower bound when employing generic sparse solver algorithms.

**Number of collocation points** We study the effect of the number of collocation points on the 3 second rollout prediction error by testing the MGN, MGN+OSC, and MGN+OSC+Adaptive models. The MGN is directly trained with the whole domain data.

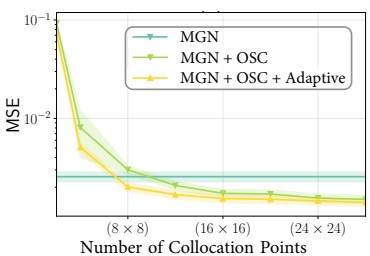

We use a different number of collocation points (from $(2 \times 2)$ to $(28 \times 28)$) in the MPNN process in the rest two models and then compare the output of the OSC with the whole domain to train. With the increase in the number of collocation points, Fig. 5.3 shows that the MGN+OSC and MGN+OSC+Adaptive achieve significant improvements in prediction accuracy over the MGN. The MGN+OSC+Adaptive has a stable better performance than the MGN+OSC, and the improvement is larger when there are fewer collocation points. The reason is that with fewer collocation points, the MGN+OSC has insufficient ability to learn the whole domain. With the adaptive collocation point, the MGN+OSC+Adaptive can focus on hard-to-learning regions during training to obtain overall better predictions.

Figure 5.3: MSE vs the number of collocation points. MGN does not use collocation points, hence constant. The more the collocation points, the better the performance.

**Number of rollout steps** We show the effectiveness of the OSC method in improving long-range prediction accuracy by comparing the MGN and MGN+OSC model. Fig. 5.4 shows the MGN+OSC can keep stable in long-range rollouts compare with the MGN.

The reason is that with the OSC, we can use fewer neural network rollout steps to obtain longer-range predictions, which avoids error accumulation during the multi-step rollouts and implicitly learns for compensating integration residual errors. In addition, end-to-end learning lets the neural networks in MGN+OSC learn the states between rollout steps, making the prediction stable and accurate.

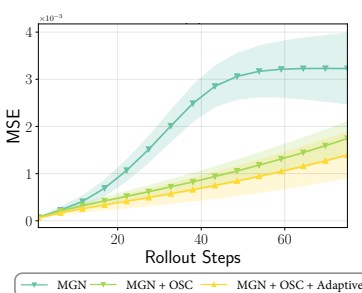

Figure 5.4: MSE vs rollout steps. Our full model yields stable long rollouts.

**Number of Message Passing Layers** We further showcase the sensitivity to the number of message-passing layers while adopting the same baseline processor for a fair comparison. As Fig. 5.5 shows, a higher number of message-passing layers generally helps in decreasing the MSE at the cost of higher runtime, reflecting findings from prior work such as GNS (Alet et al., 2019) and MGN (Pfaff et al., 2021). Notably, GRAPHSPLINENETS exhibits a Pareto-optimal trade-off curve, outperforming MGN in terms of prediction accuracy and computational efficiency at different numbers of message-passing layers. Downsampling mesh nodes can help in robustness and generalization given the larger spatial extent of communication akin to Fortunato et al. (2022); the OSC plays a crucial part in handling the problem's continuity (e.g., in time) while adaptive collocation effectively helps the model to focus on regions with higher errors thus increasing the model performance.

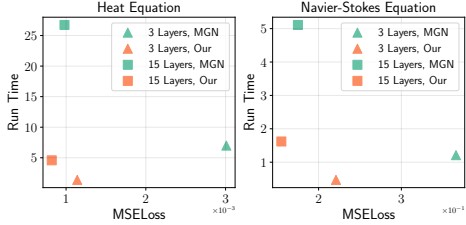

Figure 5.5: Pareto plot for MSE and runtime at different # of layers.

**Scalability to Higher Dimensions** We showcase GRAPHSPLINENETS on a larger 3D dataset, namely the 3D Navier-Stokes equations dataset from PDEBench (Takamoto et al., 2022) (more details in Appendix B.7). The number of nodes in Graph-SplineNets is 4096, which is more than $4\times$ compared to the Black Sea Dataset. Moreover, the 3D Navier-Stokes includes a total of 5 variables: density, pressure, and velocities on 3 axes. For a fair comparison, we keep the model structure the same as in other experiments and compare GRAPHSPLINENETS against MGN. MGN obtain an MSE of $(4.52 \pm 0.23) \times 10^{-1}$ with a runtime of more than 80 seconds, while GRAPHSPLINENETS remarkably outperforms MGN with an MSE of $(3.98 \pm 0.27) \times 10^{-1}$ and less than 9 seconds to generate a trajectory - an example of which is shown in Fig. 5.6.

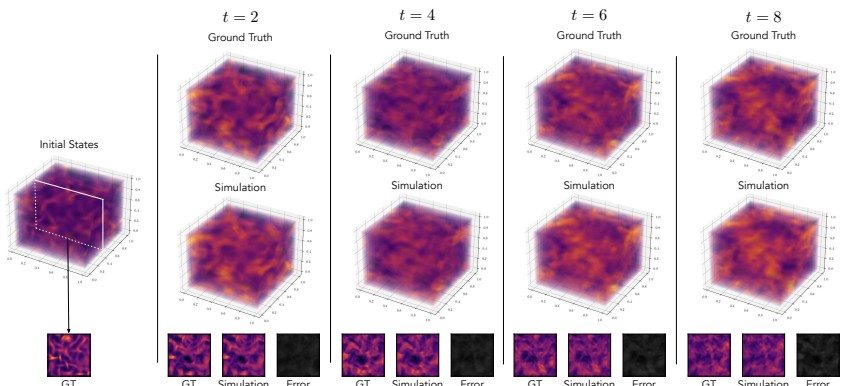

Figure 5.6: Rollout visualization of density for 3D Navier-Stokes equations.

## 5.5 Qualitative analysis

We visualize the prediction of our model with baselines on the Black Sea dataset Fig. 5.7. Our model can accurately predict even with such complex dynamics from real-world data that result in turbulence phenomena while considerably cutting down the computational costs as shown in Table 5.1. We also provide series of qualitative visualizations with different datasets and baselines in Appendix C. Our model has a smoother error distribution and more stable long-range prediction. Thanks to the continuous predictions from GRAPHSPLINENETS, we can simulate high resolutions without needing additional expensive model inference routines, while the other two models can only achieve lower-resolution predictions. Baselines visibly accumulate errors for long-range forecasts; our model can lower the error with smoother and more accurate predictions in space and time.

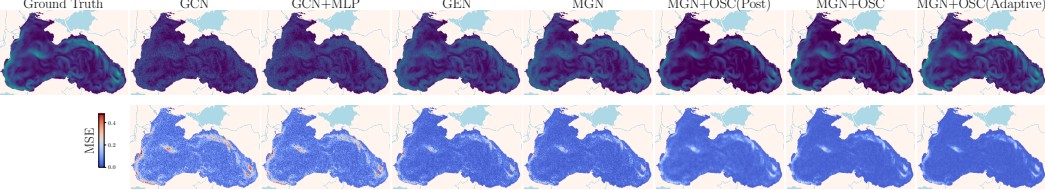

Figure 5.7: Visualization of rollout forecasts for baselines and GRAPHSPLINENETS on the Black Sea dataset. First row: predictions of the wind speed norm. Second row: mean square error (MSE). Our full model, namely MGN+OSC(Adaptive), generates more accurate predictions than other baseline surrogate models.

## 6 Conclusion and Limitations

In this work, we proposed GRAPHSPLINENETS, a novel approach that employs graph neural networks alongside the orthogonal spline collocation method to enable fast and accurate forecasting of continuous physical processes. We used GNNs to obtain predictions based on coarse spatial and temporal grids, and the OSC method to produce predictions at any location in space and time. Our approach demonstrated the ability to generate high-resolution predictions faster than previous methods, even without explicit prior knowledge of the underlying differential equations. Additionally, we proposed an adaptive collocation sampling strategy that improves the prediction accuracy as the system evolves. We demonstrate how GRAPHSPLINENETS are robust in predicting complex physics in both simulated and real-world data. We believe this work represents a step forward in a new direction in the research area at the intersection of deep learning and dynamical systems that aims at finding fast and accurate learned surrogate models.

A limitation of our approach is that, by utilizing the orthogonal spline collocation (OSC) method for interpolation, we assume some degree of smoothness in the underlying physical processes between collocation points; as such, our approach may struggle when modeling discontinuity points, which would be smoothed-out. Finally, we do note that despite benefits, deep learning models including GRAPHSPLINENETS often lack interpretability. Understanding how the model arrives at its predictions or explaining the underlying physics in a transparent manner can be challenging, hindering the model's adoption in domains where interpretability is crucial.

## Acknowledgements

We want to express our gratitude towards the anonymous reviewers who greatly helped us improve our paper. This work was supported by the Institute of Information & communications Technology Planning & Evaluation (IITP) grant funded by the Korean government(MSIT)(2022-0-01032, Development of Collective Collaboration Intelligence Framework for Internet of Autonomous Things).

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

# Learning Efficient Surrogate Dynamic Models with Graph Spline Networks
## *Supplementary Material*

## Table of Contents

## A   Additional OSC Material

We further illustrate the OSC method by providing numerical examples in this section.

### A.1   1-D OSC Example

For simplicity and without loss of generality, we consider the function domain as unit domain $[0,1]$ and we set $N = 3, r = 2$, which means we will use a three-order three-piece function to simulate the 1-D ODE problem. We firstly choose the partition points as $x_i, i = 0, \cdots, 3, x_0 = 0, x_3 = 1$. The number of partition points is $N + 1 = 4$. Distance between partition points can be fixed or not fixed. Then, based on the Gauss-Legendre quadrature rule, we choose the collocation points. The number of collocation point within one partition is $r - 1 = 1$, so we have in total $N \times (r - 1) = 3$ collocation points $\xi_i, i = 0, \cdots, 3$.

After getting partition points and collocation points, we will construct the simulator. Here we have three partitions; in each partition, we assign a 2 order polynomial

$$a_{0,0} + a_{0,1}x + a_{0,2}x^2, x \in [x_0, x_1] \tag{4a}$$

$$a_{1,0} + a_{1,1}x + a_{1,2}x^2, x \in [x_1, x_2] \tag{4b}$$

$$a_{2,0} + a_{2,1}x + a_{2,2}x^2, x \in [x_2, x_3] \tag{4c}$$

Note that these three polynomials should be $C^1$ continuous at the connecting points, i.e., partition points within the domain. For example, Eq. (4a) and Eq. (4b) should be continuous at $x_1$, then we

can get two equations

$$\begin{cases} a_{0,0} + a_{0,1}x_1 + a_{0,2}x_1^2 &= a_{1,0} + a_{1,1}x_1 + a_{1,2}x_1^2 \\ 0 + a_{0,1} + 2a_{0,2}x_1 &= 0 + a_{1,1} + 2a_{1,2}x_1 \end{cases} \tag{5}$$

For boundary condition

$$\hat{u}(x) = \begin{cases} b_1, x = x_0 \\ b_2, x = x_3 \end{cases} \tag{6}$$

we can also get two equations

$$\begin{cases} a_{0,0} + 0 + 0 &= b_1 \\ a_{1,0} + a_{1,1} + a_{1,2} &= b_2 \end{cases} \tag{7}$$

Let us summarize the equations we got so far. Firstly, our undefined polynomials have $N \times (r+1) = 9$ parameters. The $C^1$ continuous condition will create $(N-1) \times 2 = 4$ equations, and the boundary condition will create 2 equations. Then we have $N \times (r-1)$ collocation points. For each collocation point, we substitute it to polynomials to get an equation. For example, if the ODE is

$$\hat{u}(x) + \hat{u}'(x) = f(x), x \in [0, 1] \tag{8}$$

By substituting collocation point $\xi_0$ into the equation, we can get

$$\hat{u}(\xi_0) + \hat{u}'(\xi_0) = f(\xi_0)$$
$$\implies a_{0,0} + a_{0,1}\xi_0 + a_{0,2}\xi_0^2 + a_{0,1} + 2a_{0,2}\xi_0 = f(\xi_0) \tag{9}$$
$$\implies a_{0,0} + a_{0,1}(\xi_0 + 1) + a_{0,2}(\xi_0^2 + 2\xi_0) = f(\xi_0)$$

Now we can know that the number of equations can meet with the degree of freedom of polynomials

$$\underbrace{(r+1) \times N}_{\text{Parameters}} = \underbrace{2}_{\text{Boundary}} + \underbrace{(N-1) \times 2}_{C^1\text{Continuous}} + \underbrace{N \times (r-1)}_{\text{Collocation}} \tag{10}$$

In this example, generated equations will be constructed to an algebra problem $\mathbf{Aa} = \mathbf{f}$ where the weight matrix is an ABD matrix.

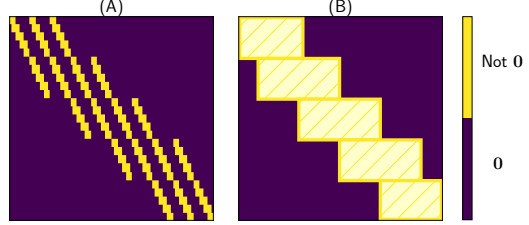

Figure A.1: Visualization of an ABD matrix.

$$\mathbf{A} = \begin{bmatrix} 1 & 0 & 0 & 0 & 0 & 0 \\ 1 & \xi_0 + 1 & \xi_0^2 + 2\xi_0 & 0 & 0 & 0 \\ 1 & x_1 & x_1^2 & -1 & -x_1 & -x_1^2 \\ 0 & 1 & 2x_1 & 0 & -1 & -2x_1 \\ 0 & 0 & 0 & 1 & \xi_1 + 1 & \xi_1^2 + 2\xi_1 \\ 0 & 0 & 0 & 1 & 1 & 1 \end{bmatrix}, \tag{11a}$$

$$\mathbf{a} = \begin{bmatrix} a_{0,0} \\ a_{0,1} \\ a_{0,2} \\ a_{1,0} \\ a_{1,1} \\ a_{1,2} \end{bmatrix}, \mathbf{f} = \begin{bmatrix} b_1 \\ f(\xi_0) \\ 0 \\ 0 \\ f(\xi_1) \\ b_2 \end{bmatrix}. \tag{11b}$$

By solving this problem, we can obtain the simulation results. Note that we can employ the COL-ROW sparse solver to solve the system efficiently (see § 3.4).

## A.2  2–D OSC Example

For simplicity and without loss of generality, we consider the function domain as unit domain $[0, 1] \times [0, 1]$, and set $N_x = N_y = 2, r = 3$. Partition points and collocation points selection is similar to the 1-D OSC method; we have $N^2 \times (r-1)^2 = 16$ collocation points in total. For simplicity, we note the partition points at two dimensions to be the same, i.e., $x_i, i = 0, 1, 2$. Unlike the 1–D OSC

method, we choose Hermite bases to describe as the simulator, which keeps $C^1$ continuous. As a case, the base function at point $x_1$ would be

$$H_1(x) = f_1(x) + g_1(x)$$

$$f_1(x) = \begin{cases} \frac{(x-x_0)(x_1-x)^2}{(x_1-x_0)^2}, x \in (x_0, x_1] \\ \frac{(x-x_2)(x-x_1)^2}{(x_2-x_1)^2}, x \in (x_1, x_2] \end{cases} \tag{12}$$

$$g_1(x) = \begin{cases} +\frac{[(x_1-x_0)+2(x_1-x_0)](x-x_0)^2}{(x_1-x_0)^3}, x \in (x_0, x_1] \\ +\frac{[(x_2-x_1)+2(x-x_1)](x_2-x)^2}{(x_2-x_1)^3}, x \in (x_1, x_2] \end{cases}$$

We separately assign parameters to basis functions, i.e. $H_1(x) = a_{1,i}f_1(x)+b_{1,i}g_1(x)$ for $x$ variable in $[x_0, x_1] \times [y_{i-1}, y_i]$ partition. Then the polynomial in a partition is the multiple combinations of base functions of two dimensions. For example, the polynomial in the partition $[x_0, x_1] \times [y_0, y_1]$ is

$$[a^x_{0,1}f_0(x) + b^x_{0,1}g_0(x) + a^x_{1,1}f_1(x) + b^x_{1,1}g_1(x)]$$
$$\times[a^y_{0,1}f_0(y) + b^y_{0,1}g_0(y) + a^y_{1,1}f_1(y) + b^y_{1,1}g_1(y)] \tag{13}$$

Now we consider the freedom degree of these polynomials. From definition, we have $2n(r-1)(n+1) = 24$ parameter. Considering boundary conditions, we have $24 - 4 \times N = 16$ parameters. The number is equal with collocation points $N^2 \times (r-1)^2$, which means we can get an algebra equation by substituting collocation points. Solving this equation, we can get the simulator parameters.

We can similarly have multiple basis functions and set parameters to the simulation result for the higher dimension OSC method. And then, select partition points and collocation points using the same strategy as the 2-D OSC method. The rest algebra equation generating and solving equations parts will not be different.

### A.3 Simple Numerical Example

We set $N = 3, r = 3$ to simulate the problem

$$\begin{cases} u + u' = \sin(2\pi x) + 2\pi\cos(2\pi x) \\ u(0) = 0 \\ u(1) = 0 \end{cases} \tag{14}$$

we can get a simulation solution as follows, which is visualized in Fig. A.2.

$$\hat{u}(x) = \begin{cases} 6.2x - 0.4x^2 - 31.4x^3, x \in [0, 1/3) \\ 1.5 + 1.6x - 13.8x^2 + 9x^3, x \in [1/3, 2/3) \\ 28.5 - 100x + 108.5x^2 - 37x^3, x \in [2/3, 1] \end{cases} \tag{15}$$

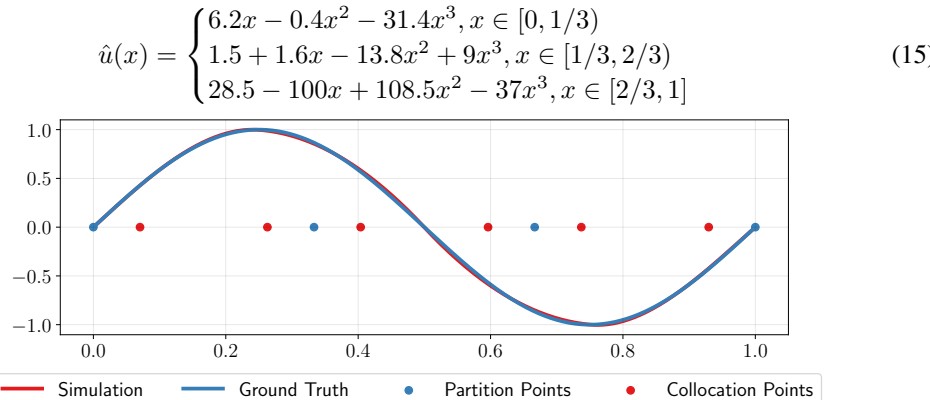

Figure A.2: Visualization of an OSC solution.

### A.4 Numerical Analysis for interpolation and collocation methods

We compared the OSC with linear, bilinear, 0–D cubic, and 2–D cubic interpolation methods on four types of problems: 1–D linear, 1–D non-linear, 2–D linear, and 2–D non-linear problems. In these experiments, we tested different simulator orders of the OSC method. For example, we set the order of the simulator to 4 for 1–D linear problem and 2 for 2–D linear problem. When the order of the simulator matches the polynomial order of the real solution, OSC can directly find the real solution. For non-linear problems, increasing the order of the simulator would be an ideal way to get lower

loss. For example, we set the order of the simulator to $4$ for 1–D non-linear problem and $5$ for 2–D non-linear problem. Thanks to the efficient calculation of OSC, even though we use higher–order polynomials to simulate, we use less running time to get results.

Table A.1: Error of OSC and four interpolation methods on different PDEs problems: $u(x) = x^4 - 2x^3 + 1.16x^2 - 0.16x$ (1-D linear), $u(x) = \sin(3\pi x)$ (1-D non-linear), $u(x, y) = x^2 y^2 - x^2 y - xy^2 + xy$ (2-D linear), $u(x, y) = \sin(3\pi x)\sin(3\pi y)$ (2-D non-linear).

| MODEL | 1-D LINEAR | 1-D NON-LINEAR | 2-D LINEAR | 2-D NON-LINEAR |
|---|---|---|---|---|
| NEAREST INTERPOLATION | $2.3670 \times 10^{-6}$ | $1.7558 \times 10^{-2}$ | $1.9882 \times 10^{-3}$ | $3.8695 \times 10^{-2}$ |
| LINEAR INTERPOLATION | $1.8928 \times 10^{-7}$ | $8.7731 \times 10^{-4}$ | $3.4317 \times 10^{-4}$ | $1.1934 \times 10^{-2}$ |
| CUBIC INTERPOLATION | $3.5232 \times 10^{-12}$ | $2.2654 \times 10^{-7}$ | $2.9117 \times 10^{-4}$ | $4.5441 \times 10^{-3}$ |
| OSC | $\mathbf{3.4153 \times 10^{-31}}$ | $\mathbf{4.1948 \times 10^{-8}}$ | $\mathbf{1.7239 \times 10^{-32}}$ | $\mathbf{3.4462 \times 10^{-5}}$ |

# B Additional Experimental Details

## B.1 Models and implementation details

In this section we introduce additional experimental details.

**Model** Specific details of model components are introduced below:

| Component | Description | Configuration |
|-----------|-------------|---------------|
| *MGN encoder* | three-layer MLP | hidden size= 64 |
| *MGN processor* | in total 3 processors, each three-layer MLP | hidden size= 64 |
| *MGN decoder* | three-layer MLP | hidden size= 64 |

Table B.1: Model Components and Configurations.

All MLPs have ReLU : $x \rightarrow \max(0, x)$ nonlinearities between layers.

Specific details of applying the OSC method are introduced as follows:

| Configuration | *Time–oriented OSC* | *Space–oriented OSC* |
|---------------|---------------------|----------------------|
| Polynomial Order | 3 | 3 |
| Collocation Points | 2 | 2 |

Table B.2: OSC Configurations.

We note that the number of collocation points in one partition is the same for each spatial dimension; features also share collocation points (i.e., wind speeds and temperature measurements share the same partition).

Regarding the choice of collocation points, for regular boundary problems, we initialize collocation points using Gauss–Legendre quadrature, following the recommendation by Bialecki (1998)[7]. This choice is particularly apt for smooth and well-behaved functions over the integration interval, making it suitable for achieving accurate results without oscillatory behavior. In irregular boundary scenarios, we employ mesh points as collocation points. Importantly, the space-oriented OSC technique enables us to obtain values at any spatial position, rendering the entire space available for collocation points, which makes the adaptive collocation points possible.

**Training** A batch size of 32 was used for all experiments, and the models were trained for up to 5000 epochs with early stopping. We used the Adam optimizer (Kingma and Ba, 2014) with an initial learning rate of 0.001 and a step scheduler with a 0.85 decay rate every 500 epoch. For all datasets, we used the split of $5 : 1 : 1$ for training, validating, and testing for a fair comparison. We set the parameter $\beta$ for adaptive OSC as constant. In particular, we set $\beta = \frac{1}{2} \times \min$ partitions width. This ensures that collocations will not shift by more than half of the partition range in one adaptation step, lending stability and robustness to the adaptive mesh.

**Space and Time Resolutions** We additionally add a more detailed overview of the experiments regarding data setup, especially for space and time resolution, such as total rollout time, space, and time resolution for both OSC space and time in Table B.3.

Note that the rollout time and the other times $\Delta t$ refer to the original dataset prediction horizons. For instance, in the Black Sea dataset, the ground truth data is provided every day (1 d), while the prediction horizon consists of 10 steps (i.e., a total of 10 days). For datasets including more than a single scalar value (Ocean Currents and Black Sea have 3 features, while 3D Navier-Stokes have 5), we the OSC is applied feature-wise.

---

[7]Note that other choices may be made. For instance, Chebyshev grids may offer advantages such as spectral convergence properties and the ability to handle function singularities more effectively. This can be an interesting avenue for future research.

| Experiment | Total Rollout Time | Step ($\Delta t$) | | Resolution | |
| | | Ground Truth | Collocation | Ground Truth | Collocation Points |
| --- | --- | --- | --- | --- | --- |
| Heat Equation | 5 s | 0.1 s | 0.2 s | $64 \times 64$ | $12 \times 12$ |
| Damped Wave | 10 s | 1 s | 2 s | $64 \times 64$ | $32 \times 32$ |
| 2D Navier-Stokes | 10 s | 1 s | 2 s | $64 \times 64$ | $32 \times 32$ |
| Ocean Currents | 10 h | 1 h | 2 h | $73 \times 73$ | $36 \times 36$ |
| Black Sea | 10 d | 1 d | 2 d | 5000 nodes | 1000 nodes |
| 3D Navier-Stokes | 0.5 s | 0.05 s | 0.1 s | 262,144 nodes | 32,768 nodes |

Table B.3: Experiment details and parameters for each dataset.

**Example Rollout**   To further elucidate the rollout procedure with GRAPHSPLINENETS, we showcase a simple example. For the 2D Navier-Stokes experiment, the ground truth trajectories consist of 10-time steps (10 seconds), with a step size of $\Delta t = 1$ second. In our model, the neural network autoregressively performs the rollout 5 times with $\Delta t = 2$ seconds, and the OSC (Orthogonal Spline Collocation) method is used to interpolate through the skipped steps. In other words, the ground truth consists of

$$Y_t, \quad t \in \{1, 2, 3, 4, 5, 6, 7, 8, 9, 10\},$$

while the neural networks in our model provide the interpolated outputs as

$$\tilde{Y}_{t_{\text{interpolate}}}, \quad t_{\text{interpolate}} \in \{2, 4, 6, 8, 10\},$$

which are then further refined using the Time-Oriented OSC to produce

$$\hat{Y}_t = \text{TimeOrientedOSC}(\tilde{Y}_{t_{\text{interpolate}}}).$$

### B.2   Heat Equation

The heat equation describes the diffusive process of heat conveyance and can be defined by

$$\frac{\partial u}{\partial t} = \Delta u \tag{16}$$

where $u$ denotes the solution to the equation, and $\Delta$ is the Laplacian operator over the domain. In a $n$-dimensional space, it can be written as:

$$\Delta u = \sum_{i=1}^{n} \frac{\partial^2 u}{\partial x_i^2} \tag{17}$$

**Dataset generation** We employ FEniCS (Logg et al., 2012) to generate a mesh from the domain and solve the heat equation on these points. The graph neural network then uses the mesh for training.

### B.3   Wave Equation

The damped wave equation can be defined by

$$\frac{\partial^2 w}{\partial t^2} + k\frac{\partial w}{\partial t} - c^2 \Delta w = 0$$

where $c$ is the wave speed and $k$ is the damping coefficient. The state is $X = (w, \frac{\partial w}{\partial t})$.

**Data generation**   We consider a spatial domain $\Omega$ represented as a $64 \times 64$ grid and discretize the Laplacian operator. $\Delta$ is implemented using a $5 \times 5$ discrete Laplace operator in simulation; null Neumann boundary condition are imposed for generation. We set $c = 330$ and $k = 50$ similarly to the original implementation in Yin et al. (2021).

## B.4 2D Incompressible Navier-Stokes

The Navier-Stokes equations describe the dynamics of incompressible flows with a 2-dimensional PDE. They can be described in vorticity form as:

$$
\begin{aligned}
\frac{\partial w}{\partial t} &= -v\nabla w + \nu\Delta w + f \\
\nabla v &= 0 \\
w &= \nabla \times v
\end{aligned}
\tag{18}
$$

where $v$ is the velocity field and $w$ is the vorticity, $\nu$ is the viscosity and $f$ is a forcing term. The domain is subject to periodic boundary conditions.

**Data generation** We generate trajectories with a temporal resolution of $\Delta t = 1$ and a time horizon of $t = 10$. We use similar settings as in Yin et al. (2021) and Kirchmeyer et al. (2022): the space is discretized on a $64 \times 64$ grid and we set $f(x, y) = 0.1(\sin(2\pi(x + y)) + \cos(2\pi(x + y)))$, where $x, y$ are coordinates on the discretized domain. We use a viscosity value $\nu = 10^{-3}$.

## B.5 Ocean Currents

The ocean currents dataset (Nardelli et al., 2013; Marullo et al., 2014) is composed of hourly data of ocean currents with an Earth grid horizontal resolution of $1/12$ degrees with regular longitude/latitude equirectangular projection[8]. We employ data from 2019-01-01 to 2022-01-01 at a depth of $0.49$ meters. We consider an area in the Pacific ocean characterized by turbulent currents corresponding to latitude and longitude (-126 $\sim$ -120, -36 $\sim$ -30). We use currents eastward and northward sea water velocities as variables, i.e., `uo` and `vo`, respectively as well as water temperature `to`.

## B.6 Black Sea

The Black Sea dataset is composed of daily real-world measurements of ocean currents and temperatures (Ciliberti et al., 2021). The resolution of the raw data is $1/27°$ x $1/36°$. We employ data starting from 01/01/2012 until 01/01/2020; we split training, validation and testing with ratios of 5:1:1 as in the other datasets in the paper sequentially on the temporal axis; i.e., so that the model has not seen data from 2018 or 2019 during training. Of the data points in the grids, less than $50\%$ actually cover the Black Sea due to the irregular shape of the sea. To obtain a mesh, we subsample the grid using Delaunay triangulation to 5000 nodes for baselines and 1000 nodes for GRAPHSPLINENETS, which results in a non-uniform mesh with an irregularly-shaped boundary akin to Lienen and Günnemann (2022). We can observe a detail of the ground truth, non-adaptive and adaptive meshing alongside GRAPHSPLINENETS errors in Figs. B.1 to B.3 respectively. We use currents eastward and northward sea water velocities as variables, i.e., `uo` and `vo`, respectively, as well as water temperature `to` at a single depth of 12.54m. We normalize features by the mean and standard deviation of the training samples. The dataset is available to download from the Copernicus Institute[9].

## B.7 3D Compressible Navier-Stokes Equations

These equations describe the dynamics of compressible fluid flows (Takamoto et al., 2022):

$$
\partial_t \rho + \nabla \cdot (\rho \mathbf{v}) = 0, \tag{19a}
$$

$$
\rho(\partial_t \mathbf{v} + \mathbf{v} \cdot \nabla \mathbf{v}) = -\nabla p + \eta\triangle \mathbf{v} + (\zeta + \eta/3)\nabla(\nabla \cdot \mathbf{v}), \tag{19b}
$$

$$
\partial_t \left[ \epsilon + \frac{\rho v^2}{2} \right] + \nabla \cdot \left[ \left( \epsilon + p + \frac{\rho v^2}{2} \right) \mathbf{v} - \mathbf{v} \cdot \sigma' \right] = 0, \tag{19c}
$$

where $\rho$ is the mass density, $\mathbf{v}$ is the velocity, $p$ is the gas pressure, $\epsilon = p/(\Gamma - 1)$ is the internal energy, $\Gamma = 5/3$, $\sigma'$ is the viscous stress tensor, $\eta$ the shear and $\zeta$ the bulk viscosities. We employ a downsampled version of the dataset from PDEBench of Takamoto et al. (2022), originally $128 \times 128 \times 128$ and subsampled to $64 \times 64 \times 64$ up to $N_t = 10$ for the ground truth data, while the

---

[8]Available at https://data.marine.copernicus.eu/product/GLOBAL_ANALYSISFORECAST_PHY_001_024
[9]Available at https://data.marine.copernicus.eu/product/BLKSEA_MULTIYEAR_PHY_007_004/description

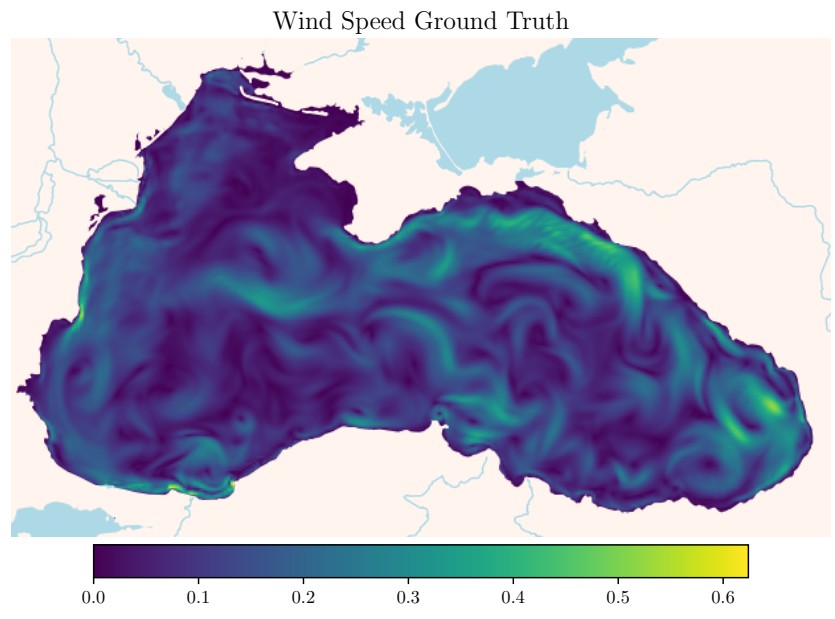

Figure B.1: Wind speed ground truth for the Black Sea dataset.

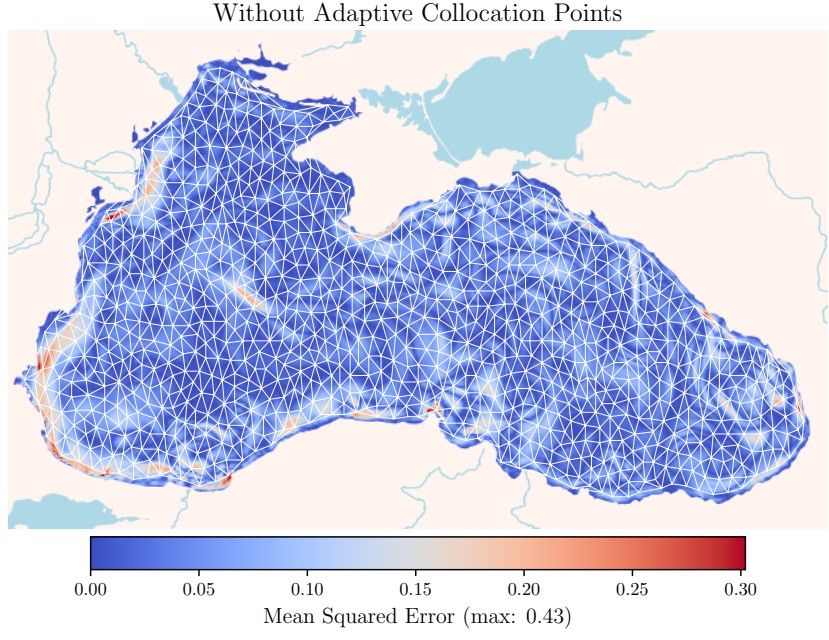

Figure B.2: MSE for GRAPHSPLINENETS predictions on the Black Sea dataset *without* adaptive meshing.

With Adaptive Collocation Points

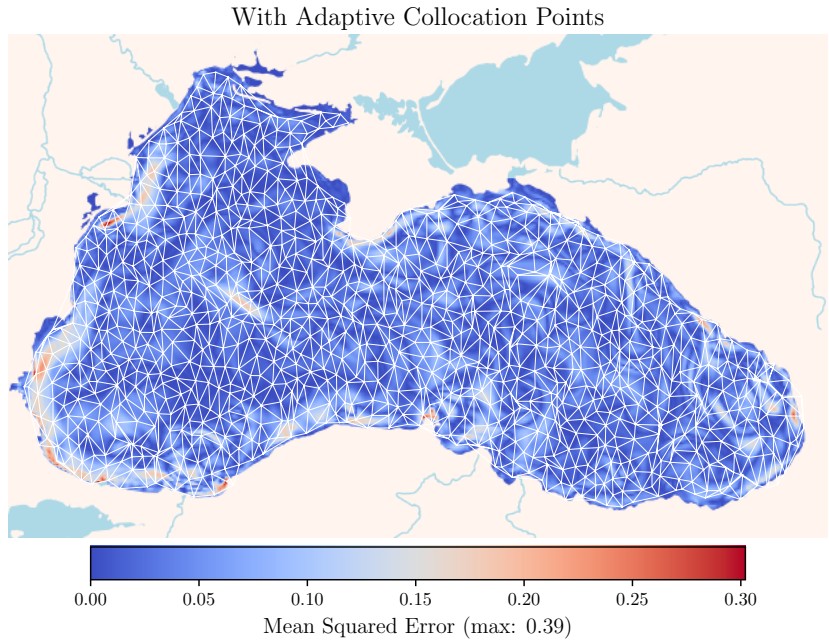

Mean Squared Error (max: 0.39)

Figure B.3: MSE for GRAPHSPLINENETS predictions *with* adaptive meshing. Adaptive meshing allows for capturing more dynamic regions better, as evidenced by the lower worst error.

collocation points lay on a dynamic grid of $32 \times 32 \times 32 = 32768$ nodes. We use a total of 600 trajectories which we divide into training and testing with a $90\%$ split as done in PDEBench.

## B.8 Hardware and Software

**Hardware** Experiments were carried out on a machine equipped with an INTEL CORE I9 7900X CPU with 20 threads and a NVIDIA RTX A5000 graphic card with 24 GB of VRAM.

**Software** Software-wise, we used FEniCS (Logg et al., 2012) for Finite Element simulations for the heat equation experiments and PyTorch (Paszke et al., 2019) for simulating the damped wave and Navier-Stokes equations. We employed the Deep Graph Library (DGL) (Wang et al., 2020) for graph neural networks. We employ as a main template for our codebase the Lightning-Hydra-Template [10] which we believe is a solid starting point for reproducible deep learning even outside of supervised learning (Berto et al., 2023); the Lightning-Hydra-Template combines the PyTorch Lightning library (Falcon et al., 2019) for efficiency with modular Hydra configurations (Yadan, 2019).

---

[10]https://github.com/ashleve/lightning-hydra-template

# C Additional Visualizations

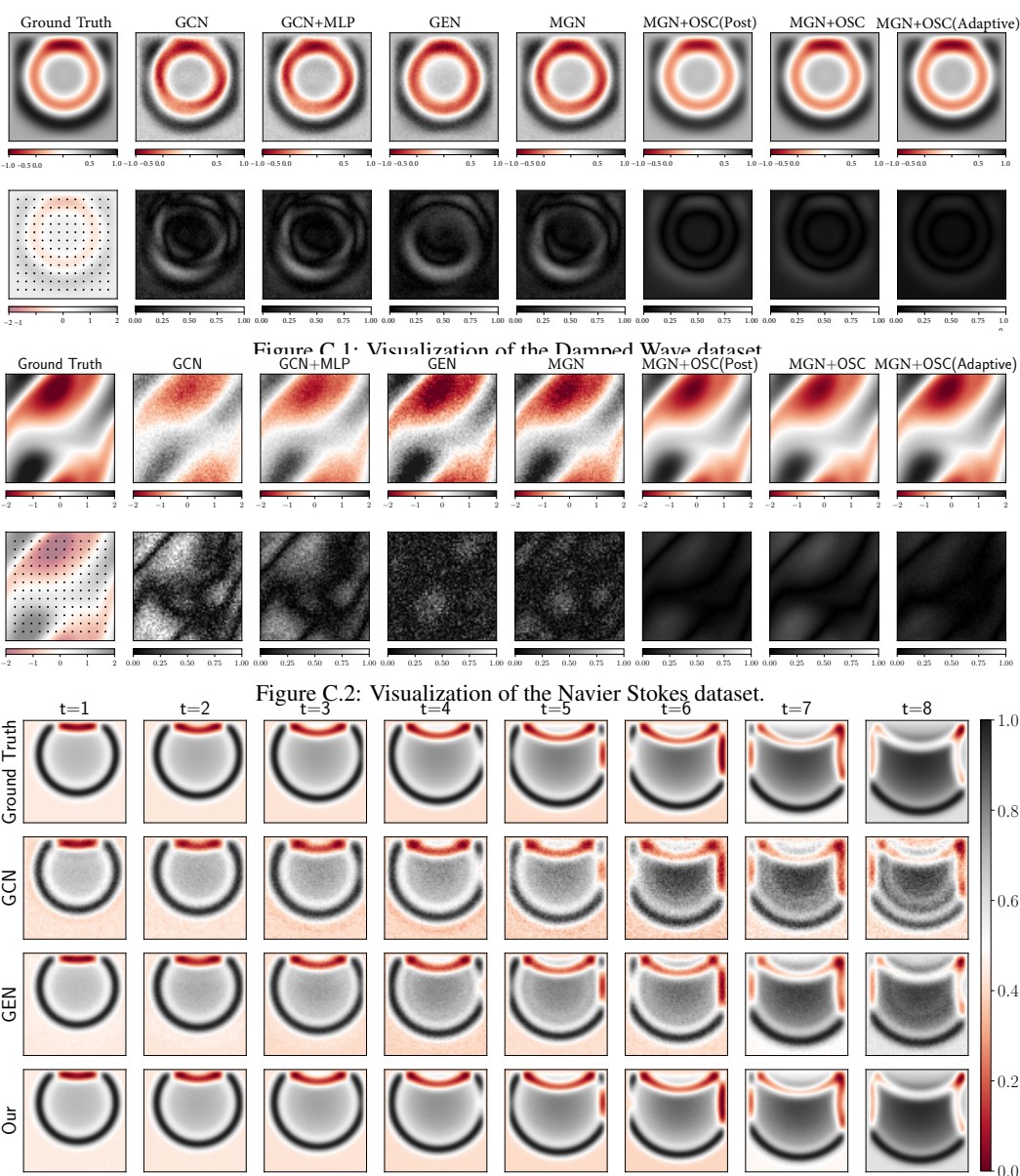

Figure C.1: Visualization of the Damped Wave dataset.

Figure C.2: Visualization of the Navier Stokes dataset.

Figure C.3: Wave dataset prediction results. GRAPHSPLINENETS generates more stable and smoother predictions compared to baselines.

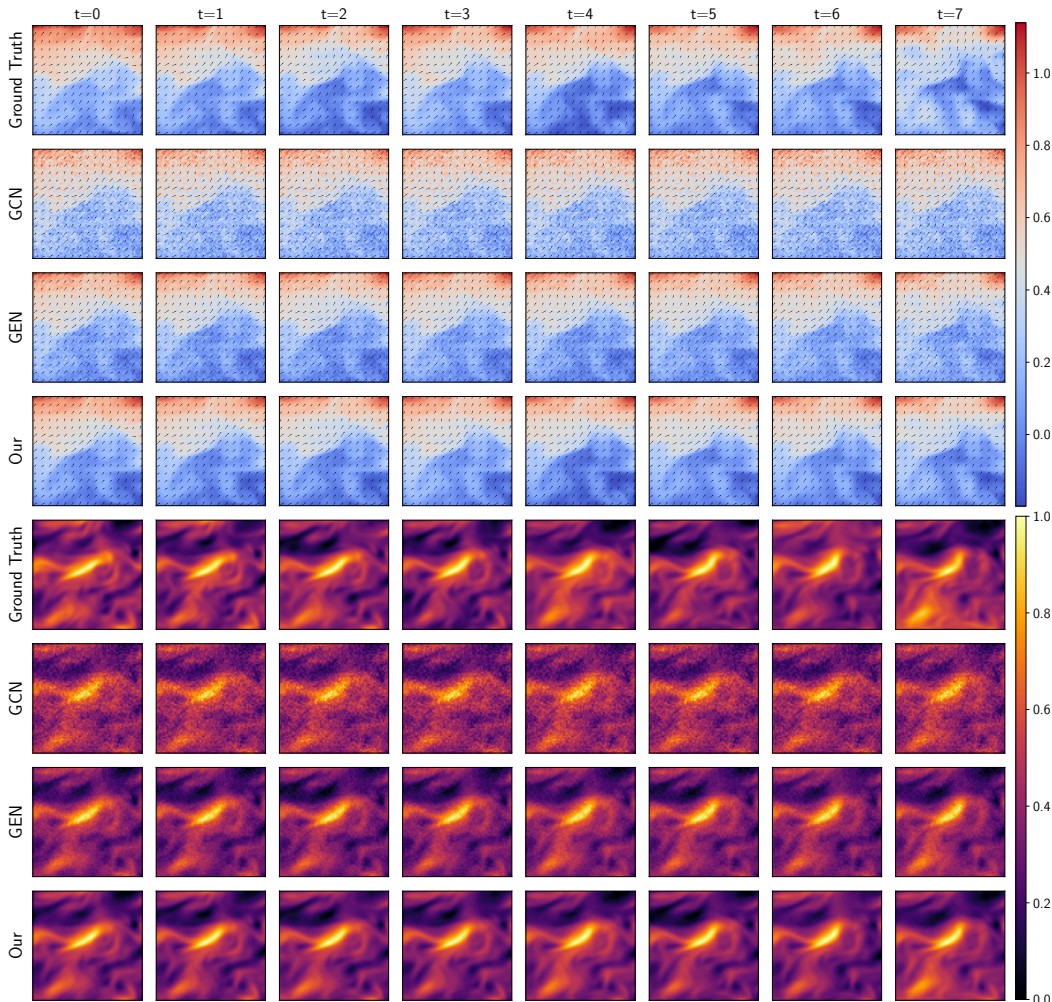

Figure C.4: Ocean dataset prediction results. GRAPHSPLINENETS generates more stable and smoother predictions.

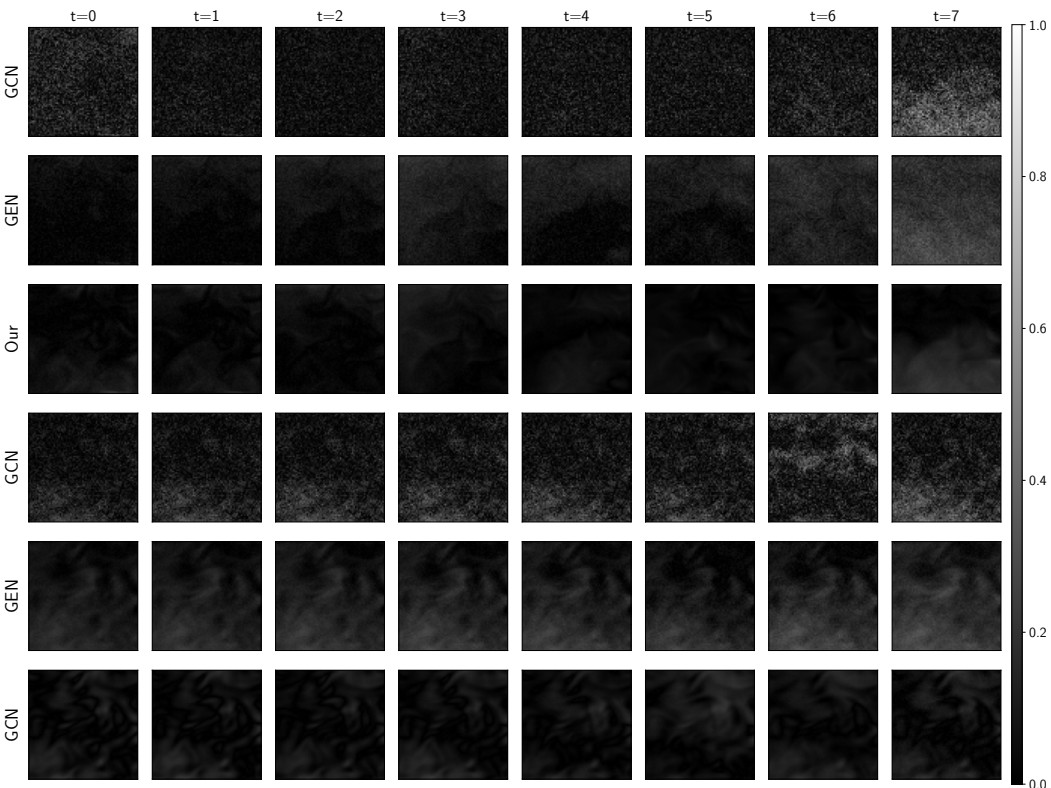

Figure C.5: Ocean dataset prediction error. GRAPHSPLINENETS contains the error better than baselines.

