# OpenReview forum: "Learning Efficient Surrogate Dynamic Models with Graph Spline Networks"
_NeurIPS.cc/2023/Conference — NeurIPS 2023 poster_

### Official Review · Reviewer_Rs2G · 2023-07-05

**Soundness:** 3 good
**Presentation:** 3 good
**Contribution:** 3 good
**Rating:** 7
**Confidence:** 3

**Summary:**

The paper builds upon a recent advances in deep-learning based collocation methods for physical simulation to improve prediction accuracy in continuous space and time. The authors proposed a hybrid model based on graph neural network and differentiable orthogonal spline collocation method. Interpolation in time as well as in space is adopted to enable employing coarser temporal grids to speed up prediction. The paper also introduces a new adaptive collocation sampling strategy and techniques for training such as not needing to learn interpolators or collocation weights to decrease the computational overhead. Experiments in various PDE settings demonstrate that the proposed model, GRAPHSPLINENETS outperforms state-of-the-art baselines in both accuracy and runtime.

**Strengths:**

The paper introduces some techniques, a novel loss function comprising OSC function, COLROW algorithm to efficiently solve linear equations associated with OSC methods, and adaptive collocation sampling to allow for prioritized sampling of important location regions. Wide variety of experiments are conducted, and the proposed method outperforms the baselines.

**Weaknesses:**

Although the authors claim that GraphSplineNets forecast continuous responses in time and space, comparison of GraphSplineNets are limited to space continuous baselines. Also, as in the main text, only a fewer points are sampled as collocation points in time — It is still difficult to assess the advantage of incorporating time-oriented OSC as well as space-oriented into the architecture. It is also unclear how is dataset for interpolation points error in the equation (2) generated.

The explanation on Figure 5.1 is too brief for me to understand. Does the size of input of MGN is larger than that of MGN + OSC (12x12)? In appearance it looks strange to me that the running time of MGN w/o OSC higher than MGN w/ OSC.


**Questions:**

* Is OSC inference time in Figure 5.1 associated with both spacial and time oriented collocation?
* Does adaptive collocation sampling also perform for important time interval?

**Limitations:**

Yes, the limitations are discussed in the main text.

---

> ### Author Rebuttal · Authors · 2023-08-10
>
> Thank you for your review and for acknowledging the strengths of our work in soundness, presentation, and contribution! We address your concerns point by point in the following sections.
>
> ### Why is collocation in time important?
>
> Your observation regarding the limited sampling of collocation points in time is on point. The incorporation of time-oriented OSC aims to facilitate predictions over extended future intervals in a single step - the reason why fewer points are sampled in time is that too large steps could eventually hurt the solutions in data-driven settings, much like it would in a normal time-stepping of an ODE via explicit methods.  By interpolating through few time steps, the advantage of our approach is twofold: it enhances computational efficiency by reducing the number of necessary prediction steps and potentially minimizes error accumulation that can arise from more frequent, smaller time-step predictions since the network can implicitly learn to compensate for time stepping error.
>
> ### How is the dataset for interpolation points error generated?
>
> We use interpolations on available data to match equation (2) errors. Therefore, the data is not generated but “given” as a dataset (more information about datasets used in Appendix B), as is common in supervised surrogate modeling.
>
> ### Runtime comparison in Figure 5.1
>
> Thanks for your question! We will clarify this in the manuscript as well. The OSC inference time in this figure is associated with both spatial and time-oriented collocation. MGN without OSC has higher runtime because MGN+OSC effectively *downsamples* the domain while still capturing the dynamics thanks to the interaction with OSC. The downsampling allows for noticeably faster runtimes and more robustness thanks to the OSC, whose overhead is minor compared to the total runtime savings.
>
> ### Adaptive Collocation in Time
>
> This is a clever idea. While we did not test it in the paper, we believe it is an exciting avenue for future work. We also have some ideas we will put down in the paper regarding this point - particularly how to handle non-regular time stepping due to the adaptive collocation in time. In this case, an additional parameter should be kept track of, namely time $\Delta t _i$ for collocation point $i$ in time; otherwise, the network would try to predict the change at a constant time step $\Delta t$ given by the base dataset. Each collocation point would have its own $\Delta t_i$ and the output of GraphSplineNets can be modulated by this variable to perform time stepping effectively. Thanks for your suggestion!

---

> > ### Comment · Reviewer_Rs2G · 2023-08-15
> > **Follow-up questions**
> >
> > Thank you for the reply.
> >
> > Can you expand the detail of MSE reported in Table 5.1 a bit more? Some of my questions on the MSE are:
> > * For each of the experiment, how many time steps do the ground truth trajectories consist of and what is the cardinality of the in-time collocation points $\\{t_{0}, t_{1}, · · · , t_{K}\\}$?
> > * On which data points is the MSE defined? Does it take the mean for all spatial points and all steps in each trajectory, and all the ground-truth trajectories in the test dataset?
> > * Is the MSE error reported in Table 5.1 comparable to the MSE error defined on only in-space collocation points over $\\{t_{0}, t_{1}, · · · , t_{K}\\}$? I’m curious if the relatively high number of data points in the ground truth dataset would have an impact to reduce the MSE defined on only in-time and in-space collocation points.
> >
> > Also, for experiment of Navier-Stokes equations, the time horizon of the dataset is t=10 (in Appendix B.3), as opposed to t=5 reported in Table 5.1. Why did you choose t=5 as the rollout length? Are there any difficulties that prevent the proposed model from predicting 10 steps?
> >
> > Is the inference time reported in Figure 5.1 evaluated by the same models that produce the results for one of the experiments in Table 5.1?

---

> > > ### Author Response · Authors · 2023-08-16
> > > **Authors response for follow-up questions**
> > >
> > > Thanks for raising these fair concerns. Let us clarify!
> > >
> > > The 5-second rollout is, in fact, our mistake and only refers to the heat equation. We use different time stepping depending on each specific dataset - for instance, the Black Sea data has a time resolution of one full day. We summarize in the following table a more detailed overview of the experiments, such as total rollout time, space, and time resolution for both OSC space and time.
> > >
> > > | Experiment | Total Rollout Time  | Ground Truth Step $\Delta t$ | Collocation  Step $\Delta t$ | Ground Truth Resolution | Collocation Points Resolution |
> > > | --- | --- | --- | --- | --- | --- |
> > > | Heat Equation | 5 Seconds | 0.1 Seconds | 0.2 Seconds | 64 * 64 | 12 * 12 |
> > > | Damped Wave | 10 Seconds | 1 Seconds | 2 Seconds | 64 * 64 | 32 * 32 |
> > > | 2D Navier Stokes | 10 Seconds | 1 Seconds | 2 Seconds | 64 * 64 | 32 * 32 |
> > > | Ocean Currents | 10 Hours | 1 Hour | 2 Hours | 73 * 73 (3 features) | 36 * 36 (3 features) |
> > > | Black Sea | 10 Days | 1 Day | 2 Days | 5000 nodes (3 features) | 1000 Nodes (3 features) |
> > > | 3D Navier Stokes | 0.5 Seconds | 0.05 Seconds | 0.1 Seconds | 262,144 nodes (5 features) | 32,768 nodes (5 features) |
> > >
> > > > For each of the experiment, how many time steps do the ground truth trajectories consist of and what is the cardinality of the in-time collocation points $\{t_0, t_1, \cdots, t_K\}$? […] For experiment of Navier-Stokes equations, the time horizon of the dataset is t=10 (in Appendix B.3), as opposed to t=5 reported in Table 5.1. Why did you choose t=5 as the rollout length? Are there any difficulties that prevent the proposed model from predicting 10 steps?
> > > >
> > >
> > > Let us take the previous table. For example, in the 2D Navier-Stokes experiment, the ground truth trajectories consist of 10-time steps (10 seconds), and step $\Delta t$ is 1 second; while in our model, the neural network autoregressively performs the rollout 5 times with $\Delta t = 2$ seconds and the OSC will interpolate through the skipped steps. In other words, the ground truth consists of $\mathbf{Y}^{t}, t\in\lbrace 1, 2, 3, 4, 5, 6, 7, 8, 9, 10\rbrace$ and the neural networks in our model gives $\mathbf{\tilde{Y}}^{t_{\mathrm{interpolate}}}, t_{\mathrm{interpolate}}\in\lbrace2, 4, 6, 8, 10\rbrace$ and interpolate to $\mathbf{\hat{Y}}^{t} = \mathrm{TimeOrientedOSC}(\mathbf{\tilde{Y}}^{t_{\mathrm{interpolate}}})$.
> > >
> > > This is similar to the space resolution, in which GraphSplineNets predict only a subset of the domain and interpolate through the rest. We will add this information to our manuscript.
> > >
> > > > On which data points is the MSE defined? Does it take the mean for all spatial points and all steps in each trajectory, and all the ground-truth trajectories in the test dataset?
> > > >
> > >
> > > The MSE is defined based on the ground truth resolution and time steps, i.e., all spacial points and all time steps in each trajectory.
> > >
> > > > Is the MSE error reported in Table 5.1 comparable to the MSE error defined on only in-space collocation points over  $\{t_0, t_1, \cdots, t_K\}$? I’m curious if the relatively high number of data points in the ground truth dataset would have an impact to reduce the MSE defined on only in-time and in-space collocation points.
> > > >
> > >
> > > As we mentioned in the previous question, the MSE reported in Table 5.1 is defined on all spacial points and all time steps in each trajectory.  We report this small experiment decoupling the effect of space and time collocation separately.
> > >
> > > |  | MGN | MGN+SpaceOSC only | MGN+TimeOSC only | MGN+SpaceOSC+TimeOSC (GraphSplineNets) |
> > > | --- | --- | --- | --- | --- |
> > > | MSE(x10^{-3}) | 3.01±0.38 | 2.87±0.54 | 1.27±0.08 | 1.14±0.11 |
> > > | Runtime [s] | 6.99±0.12 | 1.88±0.11 | 4.63±0.15 | 1.38±0.10 |
> > >
> > > The insight is that space collocation is mostly responsible for efficiency, while time collocation yields comparatively more accuracy.
> > >
> > > > Is the inference time reported in Figure 5.1 evaluated by the same models that produce the results for one of the experiments in Table 5.1?
> > > >
> > >
> > > Correct, we use the heat equation experiment to measure the times reported in Figure 5.1, while the model structure is the same across experiments. We noticed that these trends hold in general for the experiments.
> > >
> > > Please let us know if you have further questions or concerns; we will be happy to address them!

---

> > > > ### Comment · Reviewer_Rs2G · 2023-08-17
> > > > **Good paper with solid experiments**
> > > >
> > > > Thank you for providing detailed overview of the experiments and the results on ablation study. Given the rebuttals and reviews, I raised my score. The additional reports made my understanding much clearer. It’s also very interesting to see that incorporating just one of SpaceOSC and TimeOSC into MGN improves performance of the baseline MGN in both of the metrics. These results support the idea of using both space and time-oriented OSC. I think this paper is above the acceptance threshold.

---

> > > > > ### Author Response · Authors · 2023-08-17
> > > > > **Author response**
> > > > >
> > > > > Thank you for your constructive feedback and the time you have taken to understand our work. We truly appreciate your positive remarks.
> > > > >
> > > > > We noticed that while you mentioned raising the score in your comments, it may not have been updated on the system. Could you please check and confirm the score update on OpenReview? It can be done by clicking on "Edit -> Official Review" on the main review you submitted.
> > > > >
> > > > > We remain available for any further support. Thank you in advance!
> > > > >
> > > > > ---
> > > > >
> > > > > Edit: we checked it, thanks a lot!

---

### Official Review · Reviewer_jrMD · 2023-07-06

**Soundness:** 3 good
**Presentation:** 3 good
**Contribution:** 3 good
**Rating:** 7
**Confidence:** 3

**Summary:**

The authors introduce GraphSplineNets, providing a framework for combining MPNNs like MGNs with interpolation/collocation methods. Their contributions also include two methods to surpass the performance of a naive combination: end-to-end training and collocation points adaptation. Temporal and spatial downsampling allow the MPNN to run faster, while the collocation method makes possible *increased* inference-time resolution despite the NN's downsampled domain.

**Strengths:**

Significance:

1. This is an important area, and the paper's core idea is interesting and well presented (as a proof of concept, at least). The paper offers a lot of infrastructure (code and ideas) important to future steps at the intersection of GNN surrogates and collocation.
2. The inference speed benefit of GSNs is intriguing because applying GNNs at scale can be difficult.

Clarity:

3. The motivation for this work is communicated well. This work's connection to other approaches that deal with GNN scalability is clearly established.
4. The explanation of the GNN-based OSC method is very well done and nicely illustrated.

Quality:

5. OSC seems like a potentially promising approach for dealing with the efficiency challenges of GNN simulations.
6. The end-to-end optimization approach (differentiating through the OSC) is well justified and intuitive.

Originality:

7. Plugging GNNs into an OSC method, allowing temporal coarsening of MPNN (e.g. MGN) activity to increase efficiency, and optimizations of this joint GNN+OSC method (end-to-end training, adaptive points).
8. Giving automatic differentiation support to an ABD matrix equation solving routine.


**Weaknesses:**

My main concern is the quality of the evaluation of the method. I think this paper would gain significance if it showed that GSN's performance benefits hold in more challenging settings (by demonstrating why the benefits of GSN will generally be present, by evaluating on harder problems and against better tuned baselines, etc.). In other words, addressing some of the following weaknesses might help other researchers better understand the potential relevance of this method.

1. The paper leaves unclear why its methods beat baseline methods' errors (i.e., is it because downsampled mesh nodes have access to information from farther away?), if the errors it achieves are practically relevant (i.e., sufficiently low for the model to be used), and if it beats stronger baselines (i.e., MGN with more message passing layers or MultiScale MGN).

2. Moreover, no large-scale or 3D problems were considered, and this is at odds with the description of the proposed method as one that addresses efficiency challenges of existing approaches.

Minor concerns:

3. Figure 4.1 is nice, but I think at $t=0$, you should have a $y_3$ instead of a $y_0$.


**Questions:**

1. Why use only 3 MP layers in baseline MGNs? This gives the GraphSplineNets method an unfair advantage in that it can communicate over a larger spatial extent (because it is operating on a coarser mesh). Relatedly, the baseline MGN results (e.g., those in table 5.1) are rather poor compared to results shown in the MGN paper, which recommends using 15 MP layers.

2. If you keep on increasing the collocation points, will GSNs eventually provide the same solution as MGNs? Assuming this is the case, Figure 5.3 suggests that more collocation points is better, but it might be nice to see when the relationship flips and the methods converge.

3. How do GSNs perform on larger scale problems? In particular, can they compete with approaches designed to be more scalable like MultiScale MeshGraphNets (Fortunato et al., 2022)?

4. Line 27: It's true that traditional solvers can struggle, but it might make sense to support this point by citing a paper that analyzes traditional solvers rather than a paper that analyzes ML solvers.

5. In the Appendix, there seem to be some minor typos/errors in the equations. E.g., equations 7 and 11 seem to assume 2 partitions when the example has 3 partitions. This did not affect my score, but cleaning it up would be great!


**Limitations:**

The authors addressed some key limitations. I might add that no large-scale or 3D problems were considered, which hinders understanding of the benefits of the proposed method at the scales where its contribution (efficiency) might matter most.

---

> ### Author Rebuttal · Authors · 2023-08-10
>
> Thank you for the constructive and detailed review and thoughtful feedback! We provide responses to your concerns and questions below.
>
> ### Why do GraphSplineNets work well?
>
> Thank you for your question! This is an important point for our paper and as such we would like to clarify it.
>
> As you correctly noticed, we empirically found that downsampling mesh nodes can help in robustness and generalization; the OSC plays a crucial part in handling the problem’s continuity (e.g. in time) while adaptive collocation effectively helps the model to focus on some regions with higher errors.  This can be in part attributed to the fact that GraphSplineNets can communicate to a larger spatial extent; but not only, as demonstrated in our additional experiments.
>
> We include in the PDF a new figure Figure R2 and table Table R1 that show a comparison of GraphSplineNets against MGN with 3 message passing layers and 15. As we can notice from the comparison, a higher number of message-passing layers generally helps in decreasing the MSE at the cost of higher runtime, reflecting findings from prior work such as GNS[1] and Meshgraphnets[2]. For the sake of a fair comparison, the baseline processor remained consistent across all models. Notably, GraphSplineNets shows a Pareto optimal trade-off, outperforming Meshgraphnets (MGN) in terms of both prediction accuracy and computational efficiency at different numbers of message-passing layers. Given that the neural network structure is the same for both our GraphSplineNets and MGN, we attribute the increase in accuracy (and importantly, at a reduced computational cost) to our proposed method.
>
> ### Large-scale and 3D experiments
>
> We agree with you! While we focused mostly on simulated and real data from 2D domains in the main paper, given our background in fluid dynamics and weather forecasting, we do agree that large-scale 3D experiments can better demonstrate the scalability of our approach. As such, we have prepared a new experiment - namely, the 3D Navier-Stokes equations dataset from PDEBench [5]. We use the 3D Navier-Stokes dataset with a total of 600 trajectories which we divide into training and testing with a 90% split as done in PDEBench. The number of nodes in GraphSplineNets is 4096, which is more than 4 times higher than in our previous hardest experiment (the Black Sea Dataset). Moreover, the 3D Navier Stokes include a total of 5 variables: density, pressure and velocities on 3 axes. For fair comparison, we keep the model structure the same as in other experiments.
>
> We show in the attached PDF in Figure R3 and Table R2 that our GraphSplineNets can successfully scale to complex, large-scale, 3D data in both accuracy and importantly speed - compared to the MGN baseline, we were able to obtain around 10x increase in inference speed!
>
> ### Effect of increasing the number of collocation points
>
> Thanks for your question. Increasing the collocation points brings GSNs closer to the behavior of  MGNs. When the number of collocation points matches the sample points, only the time-oriented OSC works. Conversely, if the neural network in GSNs predicts every steps, only the space-oriented OSC in GSNs. In Figure 5.3, we show that more collocation points typically yield lower errors, but this will cause an increase in computational time. It's also imperative to recognize that the extended rollout accuracy in Figure 5.3 for MGN+OSC is attributed to the incorporation of the time-oriented OSC.
>
> ### Citation for traditional solvers
>
> Thanks for your suggestion. We will cite a paper analyzing traditional solvers instead. We are thinking about adding [3, 4]; however, if you have a better citation(s) in mind, we would be glad to hear more about it.
>
> ### Typos
>
> - Figure 4.1: we are impressed by your attention to detail! Thanks, we have updated the figure, which can be retrieved on the rebuttal PDF.
> - Equations 7 and 11: thanks, we will clean up the equations based on your and other reviewers feedback!
>
> ### References
>
> [1] Sanchez-Gonzalez, Alvaro, et al. "Learning to simulate complex physics with graph networks." *International conference on machine learning*. PMLR, 2020.
>
> [2] T. Pfaff, M. Fortunato, A. Sanchez-Gonzalez, and P. W. Battaglia. Learning mesh-based simulation with graph networks. *International Conference on Learning Representations*, 2021.
>
> [3] Kremer, D. M., and B. C. Hancock. "Process simulation in the pharmaceutical industry: a review of some basic physical models." *Journal of pharmaceutical sciences* 95.3 (2006): 517-529.
>
> [4] Oberkampf, William L. "Simulation accuracy, uncertainty, and predictive capability: A physical sciences perspective." *Computer Simulation Validation: Fundamental Concepts, Methodological Frameworks, and Philosophical Perspectives*(2019): 69-97.
>
> [5] Takamoto, Makoto, et al. "PDEBench: An extensive benchmark for scientific machine learning." *Advances in Neural Information Processing Systems* 35 (2022): 1596-1611.

---

> > ### Comment · Reviewer_jrMD · 2023-08-16
> > **Acknowledgement of rebuttals and reviews**
> >
> > I have read the reviews and rebuttals, and I have updated my score. Based on the rebuttal, I have some additional comments/questions that may help me advocate for this paper in the discussion phase. Regardless, given the rebuttal and original draft, I believe this paper will have high impact on the ML-for-simulation area of AI.
> >
> > Follow-up questions:
> > 1. Given that you agree that the spatial coarsening allows for communication across greater spatial extents, could you please briefly compare and contrast your method with multiscale MGN (Fortunato et al., 2022)? Such a commentary would seemingly improve your manuscript's related work section and clarify your innovation. For example, you both apply MGNs on downsampled domains, but you use different methods for coarsening, etc.
> >
> > 2. I like the Pareto frontier plots in your new figure (Figure R2)! It’s great to see that you can claim a Pareto improvement. Regarding this figure, you say, “For the sake of a fair comparison, the baseline processor remained consistent across all models.” Does this mean that all models have the same weight values, or that all models have the same processor architecture but each model has unique weights?
> >
> > 3. Thanks for adding the 3D analysis! It led me to realize that I am unsure about if and when an MGN model used the spatially and temporally downsampled domain that the GSN used. Could you please let me know if the following is correct?
> >    - MGN in Table R2 uses 3 MP steps on the original domain, and GSN uses 3 steps on the coarsened domain (leading to a large speed benefit).
> >    - The MGN method that you call "MGN+OSC(Post)" in Table 5.1 is operating on the same (spatially and temporally downsampled) domain as GSN (MGN+OSC).

---

> > > ### Author Response · Authors · 2023-08-16
> > > **Authors response for the acknowledgement and follow-up questions**
> > >
> > > Thank you for your feedback. We are thrilled to receive such a positive evaluation!
> > >
> > > We will answer your rollow-up concerns in the following sections:
> > >
> > > > 1. Given that you agree that the spatial coarsening allows for communication across greater spatial extents, could you please briefly compare and contrast your method with multiscale MGN (Fortunato et al., 2022)? Such a commentary would seemingly improve your manuscript's related work section and clarify your innovation. For example, you both apply MGNs on downsampled domains, but you use different methods for coarsening, etc.
> > >
> > > Thanks for your question! The MultiScale MeshGraphNets (MS-MGN) is an interesting work to discuss and compare against our model. Let us divide the differences into space and time, as our OSC does.
> > >
> > > ### Space
> > >
> > > Message passing is performed differently -  both methods can increase efficiency but in a different way:
> > >
> > > - MS-MGNs: they perform message passing twice, both at low and high resolution
> > > - GraphSplineNets: we perform message passing at a low resolution only and use OSC in space to obtain high resolution
> > >
> > > ### Time
> > >
> > > We also have differences in how autoregressive rollouts are done:
> > >
> > > - MS-MGNs: they use a fixed-step integrator only
> > > - GraphSplineNets: we also use a fixed-step integrator but with a coarse grid, then employ the OSC to obtain higher time resolution
> > >
> > > MS-MGNs do not employ coarser time resolution, allowing GraphSplineNets for even greater speedups.
> > >
> > > Finally, we would like to mention that we plan to extend future works GraphSplineNets with MS-MGN. We believe that GraphSplineNets can enable MS-MGN to achieve even better Pareto-fronts with multi-level hierarchical prediction!
> > >
> > > > 2. I like the Pareto frontier plots in your new figure (Figure R2)! It’s great to see that you can claim a Pareto improvement. Regarding this figure, you say, “For the sake of a fair comparison, the baseline processor remained consistent across all models.” Does this mean that all models have the same weight values, or that all models have the same processor architecture but each model has unique weights?
> > >
> > > Thank you for your positive feedback on the Pareto frontier plots in Figure R2; we're pleased that you found value in them! Regarding your question, the phrase "For the sake of a fair comparison, the baseline processor remained consistent across all models" indicates that while all models share the same processor architecture - each model is optimized separately.
> > >
> > > > 3. Thanks for adding the 3D analysis! It led me to realize that I am unsure about if and when an MGN model used the spatially and temporally downsampled domain that the GSN used. Could you please let me know if the following is correct?
> > > >     - MGN in Table R2 uses 3 MP steps on the original domain, and GSN uses 3 steps on the coarsened domain (leading to a large speed benefit).
> > > >     - The MGN method that you call "MGN+OSC(Post)" in Table 5.1 is operating on the same (spatially and temporally downsampled) domain as GSN (MGN+OSC).
> > >
> > > Certainly! We appreciate your questions and observations and will further clarify the experimental setup.
> > >
> > > Indeed, your understanding is correct:
> > >
> > > - In Table R2, the MeshGraphNet (MGN) employs 3 Message-Passing (MP) steps on the original resolution data points. On the other hand, our proposed GraphSplineNets (GSN) utilizes 3 MP steps on the collocation points, which are derived from a coarsened version of the original data. This approach contributes to the observed substantial speed improvement.
> > > - The MGN+OSC(Post) in Table 5.1 operates on the same spatially and temporally downsampled domain as GSN (MGN+OSC). Notably, the key distinction lies in the loss consideration: MGN+OSC(Post) focuses solely on the collocation points' loss since the OSC approaches are applied post-processing, whereas GSN incorporates the loss of all original resolution data points.
> > >
> > > Specific to your first statement, to make it more clear, we prepared in the following table a more detailed overview of the experiments, such as total rollout time, space, and time resolution for both OSC space and time. We will add this table to the appendix in the revised version.
> > >
> > > | Experiment | Total Rollout Time  | Ground Truth Step $\Delta t$ | Collocation  Step $\Delta t$ | Ground Truth Resolution | Collocation Points Resolution |
> > > | --- | --- | --- | --- | --- | --- |
> > > | Heat Equation | 5 Seconds | 0.1 Seconds | 0.2 Seconds | 64 * 64 | 12 * 12 |
> > > | Damped Wave | 10 Seconds | 1 Seconds | 2 Seconds | 64 * 64 | 32 * 32 |
> > > | 2D Navier Stokes | 10 Seconds | 1 Seconds | 2 Seconds | 64 * 64 | 32 * 32 |
> > > | Ocean Currents | 10 Hours | 1 Hour | 2 Hours | 73 * 73 (3 features) | 36 * 36 (3 features) |
> > > | Black Sea | 10 Days | 1 Day | 2 Days | 5000 nodes (3 features) | 1000 Nodes (3 features) |
> > > | 3D Navier Stokes | 0.5 Seconds | 0.05 Seconds | 0.1 Seconds | 262,144 nodes (5 features) | 32,768 nodes (5 features) |
> > >
> > > We will be happy to clarify additional questions and concerns if there are any!

---

### Official Review · Reviewer_3ehZ · 2023-07-07

**Soundness:** 3 good
**Presentation:** 3 good
**Contribution:** 3 good
**Rating:** 7
**Confidence:** 3

**Summary:**

The authors propose a method based on deep learning that will speed up the prediction of physical systems by reducing the size of the grid and the number of tests to be carried out. The method consists of combining a graph neural network to predict some values with two orthogonal spline collocation, one spatial and one temporal, to predict values continuously.

**Strengths:**

The paper is very clear and well written. The graphics are a great complement to the text and help to make it easier to understand.
The choice of graph neural networks seems judicious in the context of a physical system that admits links and dynamic interactions between variables. As for orthogonal spline collocation, the theoretical guarantees and computational complexity justify their use. The results show improved predictions and increased speed on various physical problems. The authors are careful to point out the limitation of their approach.

**Weaknesses:**

See limitations and questions.

**Questions:**

1) Lines 176-177: I think there is a typo, 2-D domain is written twice
2) Line 203 and equation (2): you write $L$ in the equation and then you write a rounded "L", I think you mix the notation
3) To what extent is the availability of data for physical systems not too restrictive?
4) Could we imagine replacing GNNs with physical neural networks?

**Limitations:**

The impact of GNN on the overall method needs to be quantified a little more, the level of performance required by the neural network needs to be defined. This would make it easier to adapt the method, for example by changing the GNN model for another one.
We can make the same point with OSC, trying to combine GNN with an alternative to show what the use of OSC brings specifically, in terms of runtime and performance.
This would help justify the choice of these two methods over others.

---

> ### Author Rebuttal · Authors · 2023-08-10
>
> Thank you for your review and for acknowledging the strengths of our work in soundness, presentation, and contribution! We address your concerns point by point in the following sections.
>
> ### Availability of data for physical systems
>
> Your question delves into an important consideration. The availability of data for physical systems can be task-dependent, a common trait shared with various deep-learning models. Our approach yields superior results with fewer sample points than conventional surrogate models. The adaptability of our approach contributes to the observed robustness and accuracy improvements. The extent of data availability hinges on factors such as task scale and accuracy requirements. In a practical context, our collocation points can be likened to sensors. Furthermore, the adaptive collocation points mimic movable sensors, which is a practical arrangement in which a finite number of sample points has to be adjusted to maximize the amount of information. The scalability of the system influences sensor demand—the larger the scale, the greater the need. This flexible sensor deployment addresses varying levels of data availability, offering enhanced accuracy while optimizing resource usage.
>
> ### About replacing GNNs with physical neural networks
>
> If you mean physical neural networks in the sense of adjustable hardware-based networks: we believe it can be an interesting area for future research for our method that could enable way faster runtimes! However, let us also answer in case you meant physics-informed neural networks (PINNs). In this case, the objective would be different for PINNs and GraphSplineNets: in PINNs the scope is, given an equation, we fit parameters to satisfy its conditions; in our case - deep surrogate models - we optimize parameters to fit given data, which may or may not be easily described by an equation.
>
> ### Impact of GNN discussion
>
> We appreciate your valuable feedback. The GNN plays a pivotal role as the foundation of our methodology, enabling us to effectively consider interactions between various regions while accommodating irregular meshes. It serves as a fundamental tool to capture intricate relationships and patterns within the data. For scenarios involving regular domains and boundaries, alternative models such as the Fourier Neural Operators (FNO) could be explored for space-based collocation points. Meanwhile, we indeed have flexibility in model selection for the time aspect. For instance, non-graph models could be employed, broadening the applicability of our approach.
>
> ### About typos
>
> Thanks for spotting the typos! We will fix them in the manuscript.

---

> > ### Author Response · Authors · 2023-08-19
> > **Request for Feedback**
> >
> > As the author-reviewer discussion phase is coming to an end, please let us know if there are further questions or concerns to address. Thanks in advance for your time and consideration!

---

> > > ### Comment · Reviewer_3ehZ · 2023-08-20
> > >
> > > I'd like to thank the authors for their clear answer. I'm modifying my rating, taking into account the other reviews as well. I have no further questions.

---

> > > > ### Author Response · Authors · 2023-08-21
> > > > **Thanks!**
> > > >
> > > > Thanks for increasing your evaluation score and helping us in improving our paper!

---

### Official Review · Reviewer_B9pj · 2023-07-19

**Soundness:** 3 good
**Presentation:** 1 poor
**Contribution:** 3 good
**Rating:** 6
**Confidence:** 3

**Summary:**

The presented manuscript proposes a novel architecture to predict the evolution of the dynamical system. The proposed method has two key ingredients: orthogonal spline collocation for both time and spatial dimensions and adaptive selection of the collocation points. Extensive experiments demonstrate the performance of the proposed method over alternatives in both prediction quality and runtime.

**Strengths:**

The presented method is based on the natural encoder-decoder architecture and intermediate processor via message-passing layers. The main strength of the paper is the extensive experimental evaluation and sensitivity analysis of the results to the adjusting hyperparameters. The authors consider 3 datasets generated from different types of PDEs and 2 datasets with empirical data from monitoring Ocean and black sea. Their method shows the best results in all considered tests. Moreover, the proposed approach is faster than the competitors based on graph neural networks.

**Weaknesses:**

The main weakness of the paper is its structure and presentation of the proposed method. In particular,
1) Too much attention is paid to the standard method for solving structured linear systems for computing interpolation coefficients. I suggest compressing such detailed descriptions since they are mostly textbook facts. At least a comparison with a general-purpose LU solver from PyTorch is strange since the result of such a comparison is obvious. Thus, section 3.4. can be eliminated or significantly compressed.
2) section 3.2 introduce the notation for the graph, but later there are no details on how this graph is constructed from snapshots
3) in the paragraph in lines 133-139 no information about how to select collocation points only the number of collocation points is estimated. So, it is unclear how to extract collocation points from the available samples
4) Figure 4.1 is entirely confusing. I do not find where is input, output, or ingredients of eq (1) and how they should be processed together to get the prediction. Please, specify what operations are done and in what order starting from the beginning and revise section 4.1 respectively.
5) also, please move to the main text architecture details of the encoder, decoder, and processor blocks and training setup from the supplementary materials

I will increase the score if the proper revision of the text will be made in the rebuttal phase.

**Questions:**

1) how to select the parameter $\beta$ in eq (3) ? Is it constant or adaptive too?
2) do you observe any relationship between the number of processor blocks and the resulting error?
3) why do you not consider the Chebyshev grid for collocation points and the corresponding barycentric Lagrange interpolation approach? Cheb grid should be more accurate than your static approach (Fig. 4.3 left) but less computationally intensive than the adaptive approach.
4) what item in the loss function $L_s$ or $L_i$ dominates and what are the possible explanations?

**Limitations:**

The authors provide the limitations of the proposed approach.

---

> ### Author Rebuttal · Authors · 2023-08-10
>
> Thank you for the constructive review and feedback. We provide responses to your concerns and questions below.
>
> ### Too much attention to standard methods
>
> Thank you for your suggestion. We agree with you that certain sections detailing the standard method for solving structured linear systems can be condensed. However, the emphasis on our comparison with the general-purpose LU solver from PyTorch was intentional since our aim was to highlight the novelty and efficiency of our implementation: a fully differentiable COLROW method with GPU acceleration which, to the best of our knowledge, is the first of its kind and furthers our contribution. Given its significance, especially in the context of the SciML community, we deemed it beneficial to retain this section. Nonetheless, we will refine it to make it more concise while still preserving its essence based on your suggestion.
>
> ### How is the graph constructed from snapshots?
>
> Thanks for spotting the missing explanation; we will thoroughly revise this part.  We construct the graph based on the mesh on the snapshot. In regular boundary problems, we create structured mesh based on collocation points. In irregular boundary problems, for example, in the Black Sea dataset, we use the Delaunay Triangulation to create the mesh.
>
> ### How do we select collocation points?
>
> For regular boundary problems, we initialize collocation points using Gauss–Legendre quadrature, following the recommendation by B. Bialecki [3]. This choice is particularly apt for smooth and well-behaved functions over the integration interval, making it suitable for achieving accurate results without oscillatory behavior. As outlined in Section 3.3, the requirement primarily revolves around the number of collocation points. In irregular boundary scenarios, we employ mesh points as collocation points. Importantly, the space-oriented OSC technique enables us to obtain values at any spatial position, rendering the entire space available for collocation points, which makes the adaptive collocation points to be possible. We incorporate these clarifications in Section 3.3.
>
> ### Confusing Figure 4.1 and missing architecture details
>
> Thank you for the valuable comment! We have revised the figure to depict the flow from input to output more accurately and to integrate the components of eq (1). This updated figure, available on the PDF, now offers a clearer overview of the method and elucidates the inner workings of the neural network. Based on your suggestion, we also included a new supporting figure showing the encoder-processor-decoder structure ”Neural Network (NN)” in the new figure. We will revise Section 4.1 based on your suggestions, including cleaning up the notation and better clarification on the overall process, such as inputs and outputs!
>
> ### About $\beta$
>
> $\beta$ is constant. In our experiments we set $\beta$ to be $\beta = \frac{1}{2}\times \mathrm{min\ partitions\ width}$. This ensures that in one adaptation step, collocations will not shift by more than half of the partition range, lending stability and robustness to the adaptive mesh. We will include this clarification in the appendix.
>
> ### Error and number of processor layers
>
> We conducted more experiments to showcase this trend in Table R1 and Figure R2 in the attached PDF. Generally, with an increase in processor blocks, there is a trend of decreased error at the cost of higher runtime, reflecting findings from prior work such as GNS [1] and Meshgraphnets [2]. For the sake of a fair comparison, the baseline processor remained consistent across all models. Notably, GraphSplineNets offers a Pareto optimal trade-off, outperforming Meshgraphnets (MGN) in terms of both prediction accuracy and computational efficiency.
>
> ### Using the Chebyshev grid for collocation points
>
> Your question is on point. Figure 4.3 shows a regular grid for illustrative purposes, but this is not the only possible choice. As we mentioned in the previous reply about the approach to selecting collocation points, the Gauss–Legendre quadrature is particularly for functions that exhibit smooth behavior over the interval of integration. It excels in scenarios involving finite intervals, offering accurate results without introducing oscillatory behavior. However, Chebyshev grids may offer advantages such as spectral convergence properties and the ability to handle function singularities more effectively. This can be an interesting avenue for future research.
>
> ### Effect on sample points and interpolation points losses
>
> Table 5.1 shows an ablation study on these losses. The MGN+OSC(Post) works in a post-processing way, so it doesn’t utilize the $L_i$; while the MGN+OSC uses both losses and enhances the performance. We can see that while $L_s$ works as a primary contributor to the observed improvements, $L_i$ also plays a meaningful role, enabling the model to encompass a broader spectrum within the feature space and enhance its ability to discern intricate patterns. We will clarify this in the manuscript!
>
> ### Final note
>
> Thank you for considering the adjustment of the score based on the revisions! However, as per this year's NeurIPS guidelines, modifications to the paper text during the rebuttal phase are not allowed. As such, we kindly request you evaluate our submission based on the clarifications provided in our responses above. We're grateful for your understanding and constructive feedback.
>
> ### References
>
> [1] Sanchez-Gonzalez, Alvaro, et al. "Learning to simulate complex physics with graph networks." *International conference on machine learning*. PMLR, 2020.
>
> [2] T. Pfaff, M. Fortunato, A. Sanchez-Gonzalez, and P. W. Battaglia. Learning mesh-based simulation with graph networks. *International Conference on Learning Representations*, 2021.
>
> [3] Bialecki, Bernard, and G. Fairweather. "Orthogonal spline collocation methods for partial differential equations." *Journal of Computational and Applied Mathematics* 128.1-2 (2001): 55-82.

---

> > ### Author Response · Authors · 2023-08-19
> > **Request for Feedback**
> >
> > As the author-reviewer discussion phase is coming to an end, please let us know if there are further questions or concerns to address. Thanks in advance for your time and consideration!

---

> > > ### Comment · Reviewer_B9pj · 2023-08-20
> > >
> > > Dear authors,
> > >
> > > Thanks for the thorough and detailed response! I wait for the revised version of the text upon acceptance and have changed the evaluation score.

---

> > > > ### Author Response · Authors · 2023-08-21
> > > > **Thanks!**
> > > >
> > > > Thanks for increasing your evaluation score and helping us in improving our paper!

---

### Author Rebuttal · Authors · 2023-08-10

We thank all reviewers for providing valuable and constructive feedback on our work; we sincerely appreciate the time and effort dedicated to reviewing our manuscript. We are glad to receive such an overall positive evaluation on all aspects of the submission, including soundness, presentation, and contribution. In response to each reviewer's comments, we have provided a detailed explanation and made necessary revisions to improve the quality of our work.

We additionally include three common responses to summarize three major points that came out during the rebuttal:

- **Scaling GraphSplineNets to large-scale datasets**
- **Why do GraphSplineNets work well?**
- **Improvements to our manuscript**

## Scaling GraphSplineNets to large-scale datasets

Based on reviewer jrMD comments, we tested GraphSplineNets on a large-scale 3D dataset, namely, the 3D Navier-Stokes equations dataset from PDEBench [1]. We show in the attached PDF in Figure R3 and Table R2 that our GraphSplineNets can successfully scale to complex, large-scale, 3D data outperforming the baseline in both accuracy and, importantly, speed. Compared to the MeshGraphNet [2] baseline, we obtained an almost 10x increase in inference speed, while still improving accuracy. We believe GraphSplineNets would be even more impactful for efficiently learning surrogate models in even larger-scale complex datasets.

## Why do GraphSplineNets work well?

This is an important point raised by the reviewers we would like to clarify.  As correctly noted by jrMD, a reason can partly be attributed to the fact that GraphSplineNets can effectively have a longer spatial context with few message-passing layers since we downsample the domain. As also suggested by B9pj, we conducted an additional study on the effect of the number of processors (i.e., message passing layers) for both GraphSplineNets and the baseline MeshGraphNets [2] shown in Table R1 and Figure R2 of the attached PDF. Notably, GraphSplineNets shows a Pareto optimal trade-off, outperforming MGN in prediction accuracy and computational efficiency at the same number of message-passing layers. We conclude that the performance is not only due to the number of message-passing layers but also to our proposed training strategy with both time and adaptive space OSC, as demonstrated in Table 5.1 with the three variants of our method.

## Improvements to our manuscript

Based on all reviewers’ feedback, especially B9pj, we made several improvements to the manuscript and figures. We revised the main Figure 4.1 in the PDF file. Per this year's NeurIPS guidelines, modifications to the paper text during the rebuttal phase are not allowed; thus, we will upload the updated manuscript incorporating your constructive feedback after the rebuttal.

### References

[1] Takamoto, Makoto, et al. "PDEBench: An extensive benchmark for scientific machine learning." *Advances in Neural Information Processing Systems* 35 (2022): 1596-1611.

[2] T. Pfaff, M. Fortunato, A. Sanchez-Gonzalez, and P. W. Battaglia. Learning mesh-based simulation with graph networks. *International Conference on Learning Representations*, 2021.

---

### Decision · Program_Chairs · 2023-09-21

**Decision:**

Accept (poster)

**Comment:**

The paper presents a novel architecture to predict the evolution of the dynamical system. The reviewers all agree that this is a valuable and interesting result.